# Deep learning with self-supervision and uncertainty regularization to count fish in underwater images

Penny Tarling[1]*, Mauricio Cantor[2,3,4,5,6,7]‡*, Albert Clapés[1,8]‡*, Sergio Escalera[1,8]‡*

**1** Facultat de Matemàtiques i Informàtica, Universitat de Barcelona, Barcelona, Spain, **2** Departamento de Ecologia e Zoologia, Universidade Federal de Santa Catarina, Florianópolis, SC, Brazil, **3** Department for the Ecology of Animal Societies, Max Planck Institute of Animal Behavior, Radolfzell, Baden-Württemberg, Germany, **4** Department of Fisheries, Wildlife and Conservation Sciences, Marine Mammal Institute, Oregon State University, Newport, OR, United States of America, **5** School of Animal, Plant and Environmental Sciences, Witwatersrand University, Johannesburg, Gauteng, South Africa, **6** Centro de Estudos do Mar, Universidade Federal do Paraná, Pontal do Paraná, PR, Brazil, **7** Department of Evolutionary Biology and Environmental Science, University of Zurich, Zurich, Switzerland, **8** Computer Vision Center, Universitat Autònoma de Barcelona, Barcelona, Spain

☯ These authors contributed equally to this work.
‡ MC, AC and SE co-senior authors on this work.
* ptarlita7@alumnes.ub.edu (PT); mauricio.cantor@oregonstate.edu (MC); aclapes@cvc.uab.es (AC); sergio@maia.ub.es (SE)

**Data Availability Statement:** The code to our method and trained network parameters are openly available at https://github.com/ptarling/DeepLearningFishCounting. All the images and their corresponding ground-truth density maps

## Abstract

Effective conservation actions require effective population monitoring. However, accurately counting animals in the wild to inform conservation decision-making is difficult. Monitoring populations through image sampling has made data collection cheaper, wide-reaching and less intrusive but created a need to process and analyse this data efficiently. Counting animals from such data is challenging, particularly when densely packed in noisy images. Attempting this manually is slow and expensive, while traditional computer vision methods are limited in their generalisability. Deep learning is the state-of-the-art method for many computer vision tasks, but it has yet to be properly explored to count animals. To this end, we employ deep learning, with a density-based regression approach, to count fish in low-resolution sonar images. We introduce a large dataset of sonar videos, deployed to record wild Lebranche mullet schools *(Mugil liza)*, with a subset of 500 labelled images. We utilise abundant unlabelled data in a self-supervised task to improve the supervised counting task. For the first time in this context, by introducing uncertainty quantification, we improve model training and provide an accompaning measure of prediction uncertainty for more informed biological decision-making. Finally, we demonstrate the generalisability of our proposed counting framework through testing it on a recent benchmark dataset of high-resolution annotated underwater images from varying habitats (DeepFish). From experiments on both contrasting datasets, we demonstrate our network outperforms the few other deep learning models implemented for solving this task. By providing an open-source framework along with training data, our study puts forth an efficient deep learning template for crowd counting

used in this study, along with 126 sonar video files are available at "Cantor, M. (2021). Underwater surveys of mullet schools (Mugil liza) with Adaptive Resolution Imaging Sonar (Version 1.0.0). Zenodo. http://doi.org/10.5281/zenodo.4717411".

**Funding:** This work has been partially supported by the Spanish project PID2019-105093GB-I00 (MINECO/FEDER, UE) and CERCA Programme/ Generalitat de Catalunya (https://portal.mineco. gob.es), and by ICREA under the ICREA Academia programme (https://www.icrea.cat/) awarded to S. G. The data sampling was supported by research grants from the National Geographic Society (Discovery Grant WW210R-17) and the Coordenação de Aperfeiçoamento de Pessoal de Nível Superior (CAPES Brazil; https://www.gov.br/ capes/pt-br; #88881.170254/2018-01) and Conselho Nacional de Pesquisa e Desenvolvimento Tecnológico (CNPq Brazil; https://www.gov.br/ cnpq/pt-br; #153797/2016-9) granted to M.C. M.C. is supported by The Max Planck Society via the Department for the Ecology of Animal Societies at the Max Planck Institute of Animal Behaviour (https://www.ab.mpg.de/crofoot), and grants from the CAPES-DAAD PROBRAL Research Programme (#23038.002643/2018-01; (https://www.daad.de/ en/)) and the SELA CNPq-PELD Research Program (SELA 445301/2020-1). The funders had no role in study design, data collection and analysis, decision to publish, or preparation of the manuscript.

**Competing interests:** The authors have declared that no competing interests exist.

aquatic animals thereby contributing effective methods to assess natural populations from the ever-increasing visual data.

## Introduction

While biodiversity undergoes significant, rapid changes worldwide (e.g. [1]), informed decision-making in biological conservation depends on accurate empirical data. Monitoring natural populations can not only reveal how natural systems work, but it is invaluable for detecting unexpected changes, raise awareness and inform appropriate management decisions (e.g. [2]). However, counting organisms in the wild, and particularly in underwater populations, is logistically challenging. Estimating fish abundance, for instance, is critical in face of the global trend of marine resource overexploitation (e.g. [3–5]) and has traditionally relied either on intrusive, labour-intensive and/or indirect field methods, such as tissue sampling, underwater surveys and fisheries data (e.g. [6]). Although sampling methods can be species-specific and prone to biases [7], underwater videos and images generally offer a less invasive and cost-effective way to generate large volumes of data (e.g. [8]). However, efficiently processing and accurately analysing such volumes of data is a bottleneck. While the time-consuming task of manually processing underwater images has recently been improved by computer vision models, how to improve the automation, speed and precision of estimation counts of organisms in underwater imagery still remains an open problem [9, 10]. Given their sweeping success across several real-world applications, deep learning models are in the forefront of research to assess underwater natural populations from visual data.

Monitoring aquatic biota through video and photography can be particularly difficult in habitats with limited water visibility. Sonar imaging systems are increasingly employed to sample underwater populations where visibility is a constraint because it uses sound energy, instead of light, to generate digital images (Fig 1(a)). Sonar technology therefore can expand the investigation of aquatic biota living in previously inaccessible underwater habitats, such as deep and very turbid waters (e.g. [11, 12]). However, the resolution of these images is inherently lower than that of single-lens reflex (SLR) underwater cameras and it can be difficult, even for the human eye, to distinguish between "objects" that are captured without details (Fig 1(c)). Therefore, counting aquatic animals in sonar images comes with additional challenges for both human and machine. To date, there are limited tools available to process sonar images and counting target objects in these is still problematic, not easily adaptive to user needs. (For example, *Echoview* is a commercial software package for hydroacoustic data processing [13], delivering capabilities for water-column and bottom echosounder and sonar data processing. ESP3 is an open-source MATLAB package for visualizing and processing fisheries single-beam and split-beam acoustics data [14].) In recent studies, authors using Adaptive Resolution Imaging Sonar cameras to collect vast quantities of data were hindered by the need to manually count fish in their samples [12]. The development of sophisticated, effective computer vision models to process, and count fish quickly in, underwater visual data remains in its infancy, partially due to the lack of manually annotated visual data necessary for training computer vision models.

As with other counting applications (e.g. crowd counting in surveillance footage or counting cells in microbiological imagery), earlier computer vision methods for counting fish involved hand-crafted techniques such as blob detection [15] or the manual extraction of features such as edges to be used in regression techniques [16]. Recent attempts at detecting fish

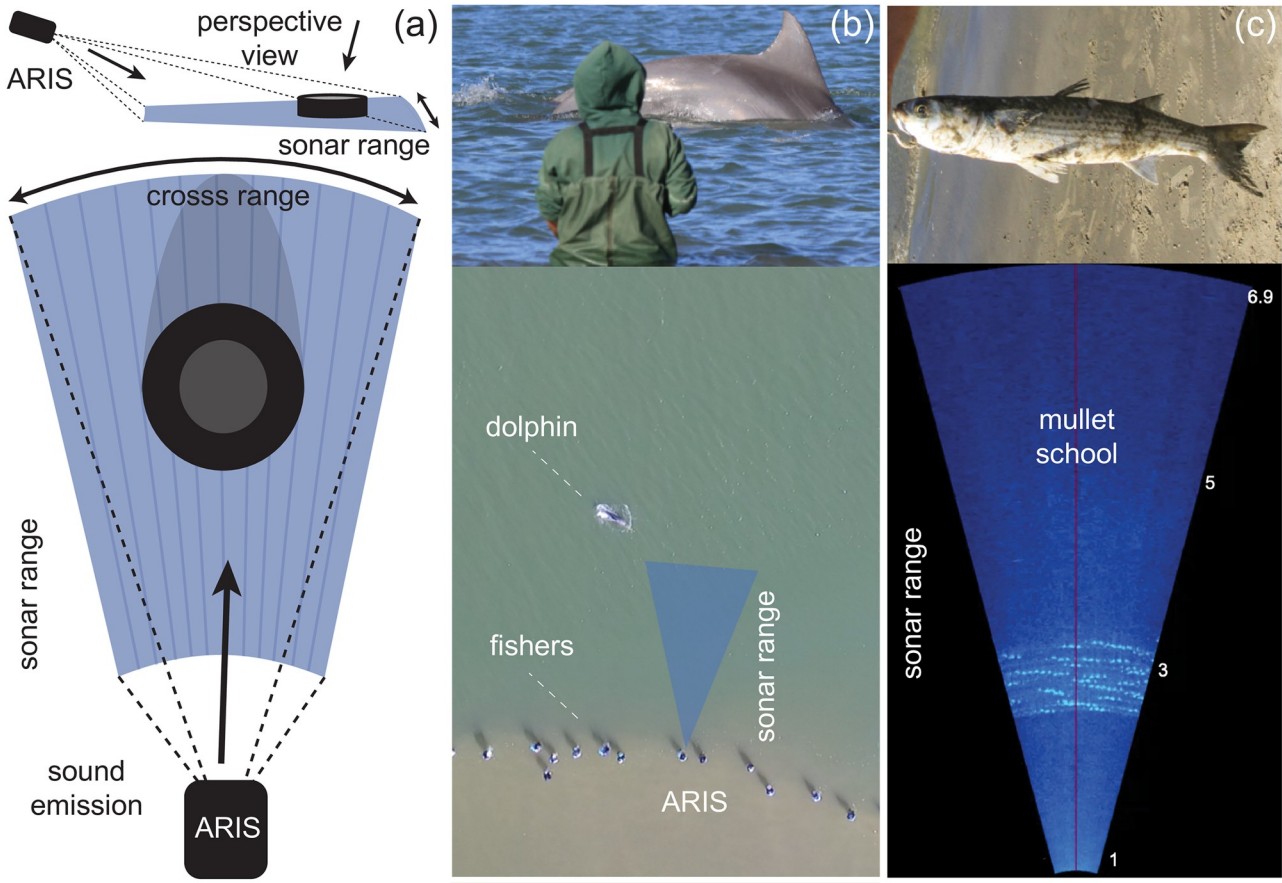

**Fig 1.** ***In situ*** **sampling of Lebranche mullet in turbid waters using a sonar imaging system.** (a) Schematics of the image production by the sonar camera. The Adaptive Resolution Imaging Sonar (ARIS) uses 128 beams to project a wedge-shaped volume of acoustic energy and convert their returning echoes into a digital image that gives an overhead view of the object, here exemplified by a cylinder (reprinted from the ARIS Explorer User Guide Manual 015045_RevB under a CC BY license, with permission from ©SoundMetrics Coorp, original copyright 2014; this modified image is similar, but not identical, to the original image, and is therefore for illustrative purposes). (b) *In situ* sonar sampling during the dolphin-fisher foraging interactions. The traditional cooperative foraging between wild dolphins and artisanal net-casting fishers targeting mullets, in the murky waters of the estuarine canal in Laguna, southern Brazil, seen from land and from a drone. Fishers wait in line at the edge of the canal for the dolphins' foraging cues (top image: a sudden dive near the coast) which fishers interpret as the moment and place to cast their nets, presumably on top of passing mullet schools. The sonar camera (blue triangle) was deployed to record passing mullet schools at the spatial scale relevant for the interacting dolphins and fishers (6–20m). (c) Lebranche mullets *(Mugil liza)*. A still image from a real-time underwater sonar video depicting the overhead perspective view of a passing mullet school in front of the line of fishers; a typical mullet caught by the fishers is shown (average body length = 42.9 cm ± 7.00 SD, n = 771 fish measured at the beach). (Photos by M. Cantor, A.M.S. Machado, D.R. Farine; reproduced with permission).

in sonar images incorporated depth-search and edge detection algorithms [17] or pixel area detection with Histogram of Gradient descriptors [18]. However, the effectiveness of the different hand-crafted approaches are bounded by the discriminative power of the manually designed set of features. Instead, deep convolutional neural networks (CNNs) are trained end-to-end, that is, the feature extraction and the learning of the meaningful patterns from those features are jointly (and automatically) optimised to solve the task at hand.

In recent years, deep CNNs have largely outperformed those more traditional approaches for object counting [19], as seen in other computer vision tasks, e.g. image classification [20] or facial recognition [21]. Deep learning therefore has potential to be the state-of-the-art solution also for underwater vision. Despite recent use of deep learning for biological monitoring in terrestrial habitats (e.g. [22, 23]), increased effort is needed for monitoring aquatic systems

[10, 24, 25]. Early use of CNNs to analyse aquatic environments include a combination of traditional detection-based and object classification methods to indirectly count organisms (e.g. [26]). More sophisticated counting approaches are gaining traction across the biological field, such as by segmentation-based methods with U-Net model architectures, where every pixel in an image is classified [27], or by object detection methods where "regions of interest" (RoI) are located and used to identify different objects usually with Faster R-CNN framework [28]. Through this methodology one can successfully detect or count animals and plants, particularly larger ones, in satellite (e.g. whales [29], elephants [30]) and aerial images (e.g. cattle [31], palm trees [32]). In underwater imaging, rather than estimating abundance directly, the focus has been on species detection and classification (e.g. [33–37]). Typically, images have been generated by SLR cameras under good underwater visibility or even constrained environments [37–39] with relatively few number of target objects (individually labelled) per image: this is the case with commonly used, publicly available benchmark datasets (e.g. Fish4Knowledge [40] & Rockfish [41]: average 1 fish/image with fish cropped and centred; DeepFish ("counting" subset): average 1 fish/image in natural habitats [42]). Under these conditions, RoI detection or segmentation based methods can work well for counting (e.g. instance segmentation with a Mask R-CNN [43]). A drawback of these methods is that expensive, time-consuming labelling is usually needed with either bounding boxes to mark the location of objects or pixel-level segmentation masks. This is particularly cumbersome for images with dense populations and it is not as an effective approach when objects are overlapping, for example when we have schools of fish.

The state-of-the-art deep learning methodology for crowd counting is a density-based approach [44]. CNNs can be trained to directly regress an image to its corresponding density map, which requires cheaper point annotations in data labelling, so has advantages over RoI detection and segmentation based methods when counting people in crowds (e.g. [45–49]) and for example overlapping cells [50]. The parallels between human crowd counting and fish —high variation in number of target objects between images, occlusions and noise—suggest these methods are efficient for counting fish in underwater images. However, there are surprisingly few examples [51, 52]. We aim to solve the task of automatic fish counting in turbid natural environments in a special natural context (Fig 1b and 1c): during the traditional fishing between artisanal net-casting fishers and wild dolphins targeting migrating mullet schools in southern Brazil (e.g. [53, 54]). These fisher-dolphin foraging interactions are thought to represent one of the few remaining cases of human-wildlife cooperation [55]. In a few estuaries in southern Brazil, wild Lahille's bottlenose dolphins *(Tursiops truncatus gephyreus)* herd migrating mullet schools *(Mugil liza)* towards the coast where a line of artisanal fishers wait for stereotyped foraging behaviours by the dolphins, which they interpret as the right moment to cast their nets [53]. Although this traditional fishing practice has been considered as mutually beneficial for both dolphins and fishers (e.g. [53, 56, 57]), the foraging benefits both predators accrued remains to be properly understood and quantified. The turbid waters complicate the estimation of the abundance of prey, thus requiring a reliable method such as the sonar-based underwater imaging system for assessing the mullet schools in the estuarine waters with very low visibility.

Here we develop an effective density-based deep learning approach to automate the process of quantifying the fish from sonar image, and provide a new dataset of manually annotated underwater images from more than 105 hours, over 1 million images, of sonar video footage recorded in this natural setting (S1 Video). The number of mullet fish between image frames can vary drastically from 0 to several hundred schooling fish, often densely compacted and difficult to distinguish between individuals. Noise in data samples (e.g. from dolphins or fishing nets) need to be correctly ignored by the automated counter. When there are biological, social

and economic consequences at stake, it is imperative the user can trust the results of an automated counter, particularly as some critics are wary of deep learning's "black-box" interpretability [58]. We combined for the first time self-supervised learning and uncertainty regularization to count fish in underwater images. Our experimental results show a marked improvement in accuracy on both our baseline deep learning model with no regularizing techniques (42% reduction in mean absolute error (MAE)), and on our experiments where a balance regularizer is incorporated, the proposed method of the only previous deep learning study to count schools of fish in low-resolution sonar images [51]. We also achieved a 21% reduction in MAE in comparison to benchmark experiments on the recent DeepFish dataset [42] containing fish recorded with traditional underwater cameras. Furthermore, our model outputs include a measure of prediction uncertainty, enabling informed biological decision making. Our model is open source and can be adopted to automate the counting of aquatic organisms in other underwater image data, as we have demonstrated on a completely different underwater dataset [42].

## Materials and methods

### Study site and data collection

Sonar-based underwater videos were recorded to quantify the availability of mullet schools (Fig 1(c)) during the cooperative foraging interactions between Lahille's bottlenose dolphins and artisanal net-casting fishers (Fig 1(b)). The water transparency at the lagoon canal is very low (from 0.3 to 1.5m visibility; collected *in situ* with a Secchi disk), mullet schools were recorded by deploying an Adaptive Resolution Imaging Sonar, ARIS 3000 (Sound Metrics Corp, WA, USA; Fig 1(a)). The videos were recorded in Laguna, southern Brazil, at the main dolphin-fisher interaction site, the Tesoura beach (28.495775 S, 48.759996 W), a 100-meter long beach at the inlet canal connecting the Laguna lagoon system to the Atlantic Ocean (e.g. [57]). The interaction site was sampled during 18 days in May-June 2018, from 09:00 to 17:00, during the peak of the mullet reproductive migration (e.g. [59]), resulting in more than 105h of video captured at 3 frames/seconds (S1 Video), totalling over 1 million images of underwater footage.

### Ethics statement

Research permits for field data sampling were obtained from the Brazilian Ministry of Environment (SISBio #47876–1, #64956–1).

### Data generation and image pre-processing

From the more than 1 million frames, 500 were selected for manual labelling. These images were chosen from 48 videos distributed across the 18 days of field sampling to encompass a wide range of variability in terms of number of fish, noise, and environmental conditions that influence water visibility. Unused (unlabelled) images could then be selected for training the proxy self-supervised task. We have made data publicly available at [60].

**Labelled data.** The videos were first manually pre-processed for contrast enhancement and background removal (Fig 2a and 2b), and then carefully chosen to include a wide range of possible sample types, from low to high fish counts and from minimal to substantial noise (Fig 2d–2f). Because our labelled dataset is small and we wanted to maximise the chance of our deep model adapting to, and test its ability on, wide-ranging and challenging observations, our dataset was not a representative sample of the field data collect.

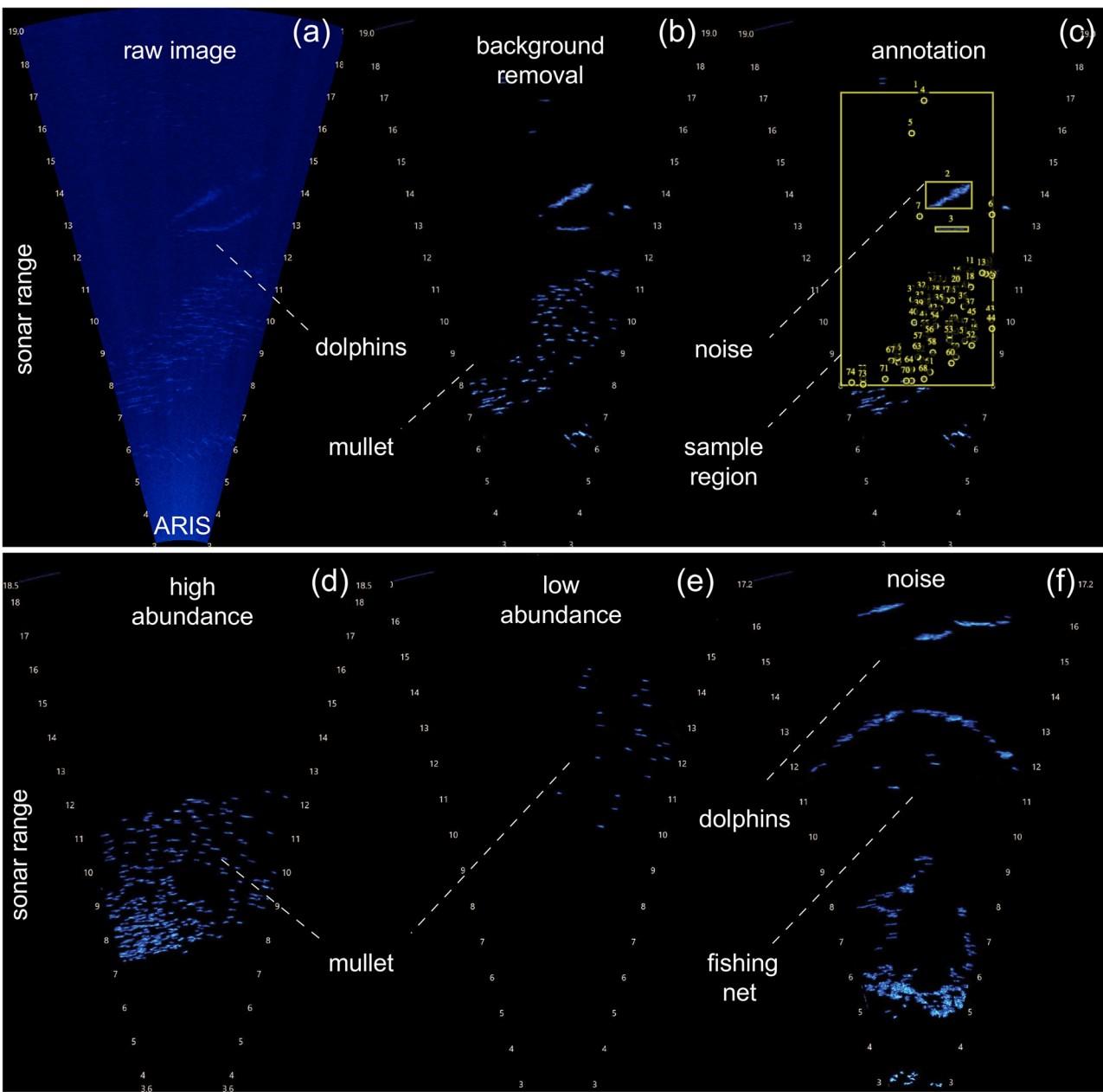

**Fig 2. Image pre-processing for assessing mullet abundance from sonar images.** (a) Raw frame depicting dolphins and a large mullet school. (b) Contrast enhancement and background removal. (c) Manual labelling of a sample: the large bounding box marks where the raw image was cropped so all input samples represent a consistent size of geographical area and at a consistent distance from the sonar camera. The smaller bounding boxes mark where noise (here, a dolphin) is present. Each point annotation marks the location of an individual mullet. (d-f) Examples of variation in the sonar images in our dataset, to which the density-based deep learning model needs to be adaptable to. (d) Frame with high mullet abundance: large number of fish, swimming compactly; (e) low abundance: small number of fish, sparsely distributed; (f) noise: 3 dolphins and a fishing net (note the overhead perspective of a rounded casting net).

As the videos taken were filmed at different ranges, the 500 images were cropped to present a geographical area of 4x8.5m$^2$, all the same distance from the camera, thus, biological population sampling will be comparable between input samples. The images were then resized to the average using bilinear interpolation, to $320 \times 576$ pixels. These images were annotated using

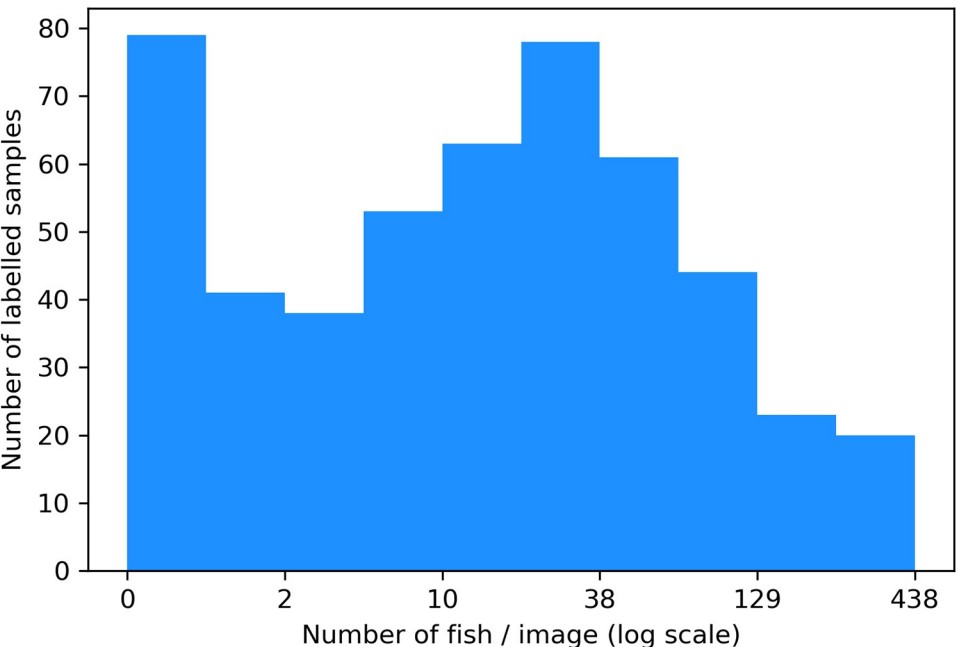

**Fig 3. Distribution of labelled dataset by number of fish.** Number of fish plotted in log scale: the subset of data is skewed towards samples with low numbers of fish. This imbalanced distribution is even more exaggerated in the complete dataset, a common theme of data collected in the wild.

the Visual Geometry Group Image Annotator [61, 62]. A point annotation was used to mark the ∼ central coordinate of a fish and a bounding box was drawn around any noise (Fig 2(c)). The point annotations of fish can then be used to derive corresponding ground truth density maps. The bounding boxes which label noise were used for subsequent data augmentation and provide opportunity for further experimental work with this dataset. Two biologists analysed a subset of samples together to reach a consensus on how to consistently identify the number of mullet, but then only one annotator annotated the whole dataset in order to avoid including different subjective biases. The abundance of mullet in a single cropped frame ranges from 0 to 438, with a mean of 42 (Fig 3). Our dataset is imbalanced with more samples containing 0 or low numbers of fish than high, typical of data collected in the wild and making the task of training deep learning models more challenging.

**Unlabelled data.**   The pool of unlabelled data contains samples with mullet in abundance of 0 to dense schools of > 500/image, and multiple sources of noise: Lahille's bottlenose dolphins, up to 4/image, fishing nets and the sea floor. Range of sonar varies between 0.7 − 10 meters (min) and 5.5 − 20.3 meters (max) and with varying camera settings affecting clarity and distortion of objects. For use in training, simple pre-processing steps were taken: images were chosen at random, cropped to the same geographical area as above and resized to 320 × 576 pixels.

## Deep learning multi-task framework

We derived an effective deep learning architecture in order to count fish to the required degree of accuracy, when labelled data is limited or costly to acquire and when images have been collected in the wild so contain noise, occlusions and in the case of sonar images, are low-resolution. Deep models trained on small datasets are prone to overfitting thus we regularize our

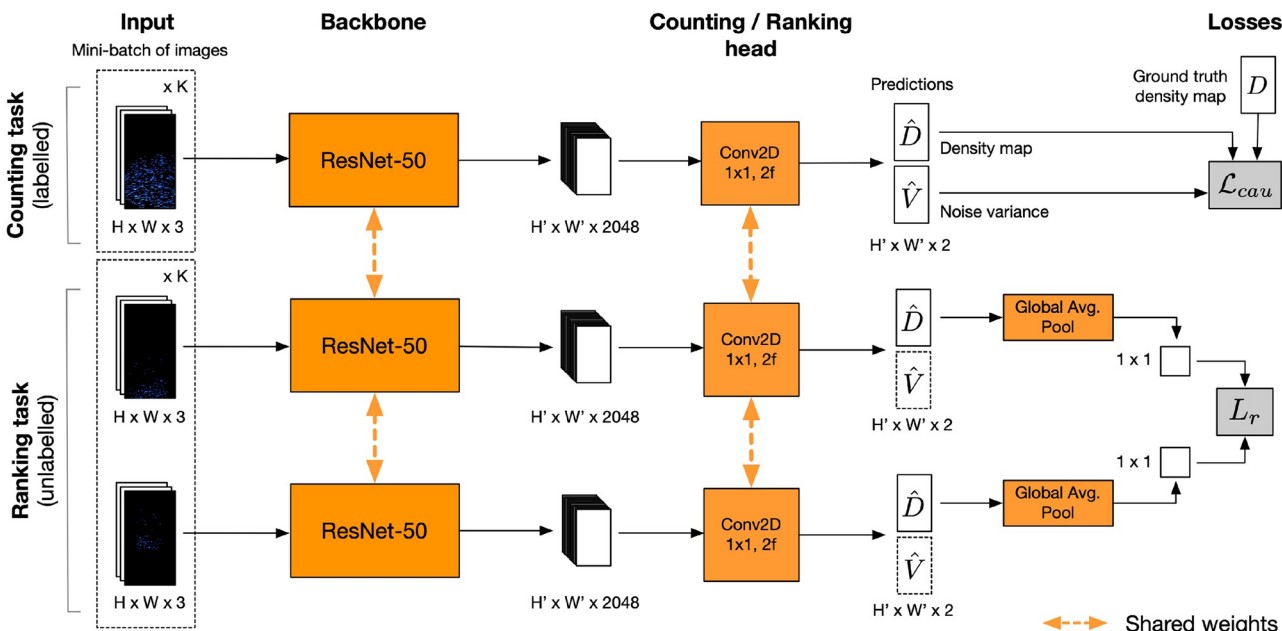

**Fig 4. Pipeline of our final network.** The multi-task network is trained end-to-end to simultaneously regress labelled images to corresponding density maps and rank the unlabelled images in order of fish abundance. The backbone of each branch is a ResNet-50 [63] followed by a $1 \times 1$ convolutional layer with 2 output filters. A non-learnable Global Average Pooling layer is added to each branch of the Siamese network so the resulting scaler count of first image in the pair $(I, I')$ can be subtracted from the second image. All parameters are shared (represented by the orange dashed arrows) thus incorporating the self-supervised task adds no parameters to the base model. The inclusion of an additional channel in our output tensor to estimate noise variance only adds a further $\sim$2x parameters in the head, equivalent to 0.01% of the total number. $K$ is the batch size, where a batch contains $K$ images from the labelled subset of data, and $K$ pairs of images from the larger unlabelled pool of data. $H$ and $W$ are the height and width of some input 3-channel RGB image, whereas $H'$ and $W'$ are the height and width of the output tensors from the backbone and heads.

learning algorithm to enhance generalisability. Our multi-task network consists of one branch that learns the supervised task of regressing an input image to an estimated fish count via a density mapping and two parallel Siamese branches which simultaneously learn the self-supervised task of ranking unlabelled images according to the number of fish. In addition, our model learns to predict the noise variance within each labelled sample. This entire framework is shown in Fig 4.

**Supervised counting task with labelled data.** As is common practice in crowd counting studies, we trained our model to regress any given image $I \in [0, 255]^{H_I \times W_I \times 3}$ to a predicted count of fish. However, following the approach of other crowd counting works [45–49, 64], instead of directly regressing the count, we predict a density map $\hat{D} \in \mathbb{R}^{H \times W}$ that can then be integrated as a proxy for the count $\hat{c}$:

$$\hat{c} = \sum_{i=0}^{H} \sum_{j=0}^{W} \hat{D}(i,j). \tag{1}$$

The backbone of our model is ResNet-50 [63], a state-of-the-art deep architecture with identity shortcut connections. Differently from the original model, the fully connected softmax layer (i.e. "FC-1000") is replaced by a counting head consisting of a 2D convolutional layer with kernel size $1 \times 1$ and 2 filters. This layer produces a tensor $\hat{Z} \in \mathbb{R}^{H \times W \times 2}$, where $H = H_I/32$ and $W = W_I/32$ are the spatial units, and 2 are the number of channels. The first channel

corresponds to the predicted density map $\hat{D}$ and the second one to the noise variance map $\hat{V} \in \mathbb{R}^{H \times W}$ that we will discuss later.

For the training of the model, the binary image masks resulting from manual point annotations (where 1-valued pixels denote presence of fish) are converted to the corresponding ground truth density maps by convolving a Gaussian kernel of size $s$ and standard deviation $\sigma$. Let $L_{\text{train}} = \{(I_i, D_i)\}, i = \{1, \ldots, N_{L_{\text{train}}}\}$ be our labelled training dataset. The model weights can then be optimised for the task of counting fish by minimising the difference between predicted and actual counts over the training data guided by a simple L1-norm absolute loss function. We refer to this as $\mathcal{L}_{\text{c}}$ to distinguish the "count" loss hereinafter. More precisely, given a batch of $K, K \leq N_{L_{\text{train}}}$, labelled images:

$$\mathcal{L}_{\text{c}} = \sum_{k=1}^{K} |c_k - \hat{c}_k|. \tag{2}$$

where $c_k = \sum_{i=0}^{H_I} \sum_{j=0}^{W_I} D_k(i,j)$ is the count integrated over the ground truth density map $D_k$, whereas $\hat{c}_k$ is the count from the predicted density map $\hat{D}$ in Eq 1 both corresponding to image $I_k$. Next, we show how we modify the loss to deal with aleatoric uncertainty introduced by different sources of noise by using the predicted noise variance map $\hat{V}$.

**Regularizing the loss term: Aleatoric uncertainty.** There are varying levels of noise within our dataset, as will be typical of data collected in an unconstrained, wild setting. There is also the challenge of manually labelling images accurately and consistently, particularly in those displaying a large number of fish. Fig 2(d) shows an example image with a dense school of fish. The fish are seen simply as blue blobs with no detailed features, making it difficult to decipher the number present when they swim close together or even overlap. Fig 2(f) shows the level of noisy objects that can occur in images that need to be distinguished from mullet. Furthermore, adjustments to camera settings and focus range affect the clarity or distortion of objects and brightness of projected image. Thus for some data, the best achievable prediction for mullet abundance, will be an uncertain prediction.

The need to model uncertainty in computer vision was highlighted by the work of [65] and these ideas were incorporated into the task of crowd counting by [66]. Aleatoric uncertainty specifically is uncertainty in the data and arises from genuine observational noise. To quantify this in regression problems, the noise variance also needs to be learned alongside the count prediction. No matter how much data we have, there will always be a degree of uncertainty in the prediction of inherently noisy images and having an understanding of this uncertainty will be invaluable. This was the motivation behind the work of [66] in crowd counting, and for our application of it to fish counting. We have focused solely on heteroscedastic aleatoric uncertainty (the assumption that observational noise varies with the input data). We hypothesise that incorporating this measure of uncertainty should not only lead to a higher level of accuracy in count predictions through optimising model training, but it will also provide the user with an understanding of the uncertainty surrounding a given result.

To do so, we estimate the noise variance of an image $I_k$, i.e. $\hat{\sigma}_k^2$, from the already produced noise variance map $\hat{V}$ as follows:

$$\hat{\sigma}_k^2 = \sum_{i=0}^{H} \sum_{j=0}^{W} \hat{V}_k(i,j). \tag{3}$$

Then we adjust our loss function to introduce this regularizer of the aleatoric uncertainty. More precisely, for a batch of $K$ images, the original counting loss $\mathcal{L}_\mathrm{c}$ now becomes $\mathcal{L}_\mathrm{cau}$:

$$\mathcal{L}_\mathrm{cau} = \sum_{k=1}^{K} \frac{|c_k - \hat{c}_k|}{\hat{\sigma}_k^2} + \log \hat{\sigma}_k^2. \tag{4}$$

There is a resulting trade off between the two components: during training, the model learns to increase the value of $\hat{\sigma}_k^2$ when the difference between $c_k$ and $\hat{c}_k$ is large to decrease its contribution to the overall loss, but minimise $\hat{\sigma}_k^2$ when the difference is small. This way, the model is able to learn to ignore noisy samples, weakening their impact on training. But, the noise variance component is added to the overall loss so the model is penalised for increasing $\hat{\sigma}_k^2$. This prevents it from simply learning to make $\hat{\sigma}_k^2$ large for all samples. Note the actual model output is $\log \hat{\sigma}_k^2$ for greater numerical stability [65] (to avoid dividing by zero). Also, as variance should be positive, this ensures the model cannot learn to make the predicted noise output negative to drive down loss: we multiply $\mathcal{L}_\mathrm{c}$ component by $\mathrm{e}^{-\,\text{"predicted noise"}}$ which would result in a large multiple if the predicted noise variance was negative. To learn this parallel prediction roughly twice the number of parameters are needed in the counting head, which is negligible compared to the overall number of parameters in the entire backbone architecture.

**Regularizing: Data augmentation of labelled data.**   We performed various augmentation techniques to increase the amount of training data. Augmentation techniques included image crops, translations, horizontal flips, small rotations, and superimposing objects (e.g. dolphin) from one image onto a randomly chosen new image. When augmenting by cropping, the crops are placed on a new blank background, similar to [67], or superimposed onto a different image. Because the sonar camera used here was tilted down so the acoustic energy beams can reach the mullets while minimizing interference from water surface reflections when generating "birds-eye view" images (Fig 1(a)), the measurable size of such large fish (Fig 1(c)) will be consistent regardless of how deep (up to ~8m) or how distant from the sonar camera (~3 to 20m) the mullets were. Scale-awareness is therefore not a key factor to consider when training the network, so the resizing of images is not used in synthetic data generation.

**Regularizing: Self-supervised ranking task with unlabelled data in a multi-task network.**   To make the most of unlabelled images and address limited availability of labelled training data, we took inspiration from the work of [64] on crowd counting. We incorporated Siamese branches in our architecture that learn to rank pairs of images according to the number of fish based on self-supervised ranking information. We can then leverage some potentially useful information on the comparative number of fish between two images, for the original counting task. Let $U = \{I_i\}, i = \{1, \ldots, N_U\}$ be the dataset of ~1M unlabelled images, hence $N_U$ being much greater than $N_{L_\mathrm{train}}$. We can then define pairs of unlabelled images, e.g. $(I_i, I_i')$, where $I_i' \in \mathcal{P}(I_i)$ is an image subregion cropped from $I_i$, thus containing equal or fewer fish. $\mathcal{P}(\cdot)$ is simply the function that generates all subregion images from $I_i$. From $U$, we generate the dataset of all possible unlabelled image pairs $P_\mathrm{train} = \{(I_i, I_i')\}, i = \{1, \ldots, N_{P_\mathrm{train}}\}$ and pick a subsample of it, namely $P_\mathrm{train}$, with a size that equates to the size of the sample of labelled image $L_\mathrm{train}$. Algorithm 1 illustrates the generation of $P$.

**Algorithm 1** Generate ranked pairs of unlabelled images

```
1: Input:
2: U = {unlabelled images}
3: Output:
4: P = {}
5: for I_i ∈ U do
```

```
 6:    x₁ = left x location of Iᵢ bounding box
 7:    y₁ = upper y location of Iᵢ bounding box
 8:    x₂ = right x location of Iᵢ bounding box
 9:    y₂ = lower y location of Iᵢ bounding box
10:    h_{I_i} = image Iᵢ height
11:    w_{I_i} = image Iᵢ width
12:    Sᵢ = []                    ▷ Output of 𝒫 function, a list of subregion
images derived from Iᵢ
13:    for f ∈ {0.25, 0.5, 0.75} do            ▷ iterate over crop
factors
14:        C_f = crop Iᵢ's region to pixels (x₁, y₁ + f * h_{I_i}, x₂ − f * w_{I_i}, y₂)
15:        choose l where 0 ≤ l ≤ f * w_{I_i}        ▷ random horizontal
translation
16:        choose u where 0 ≤ h ≤ f * h_{I_i}        ▷ random vertical
translation
17:        choose value in {0, 1} and horizontally flip if 1              ▷
random horizontal flip
18:        I_f = new blank image of size (w_{I_i}, h_{I_i})     ▷ initialize new
blank image where to place the crop
19:        place crop C_f on I_f aligning crop's topleft corner at (u, l)
               ▷ pixel location
20:        Sᵢ = Sᵢ+ [I_f]              ▷ append to the end of list
21:    end for
22: end for
23: Sᵢ = [Iᵢ] + Sᵢ               ▷ concatenate original image to the
front of list
24: for j = 1 to |Sᵢ| do            ▷ generate all combinations of
ordered pairs
25:    for k = j + 1 to |Sᵢ|
26:        P = P ∪ {(I_j, I_k)}          ▷ join sets
27:    end for
28: end for
29: return P
```

The two Siamese ranking branches share the same backbone architecture from the counting branch. The only difference is the inclusion of a global average pooling layer at the end of the ranking branches to produce the average count estimate across density maps' spatial units. This task is trained with a standard pairwise ranking hinge loss $\mathcal{L}_r$, applicable to a Siamese architecture. For a batch of $K$ (s.t. $K \leq N_{U_{\text{train}}}$) unlabelled image pairs:

$$\mathcal{L}_r = \sum_{k=1}^{K} \max(0, \ p_k{}' - p_k + \epsilon) \tag{5}$$

where $\hat{p}'_k = \frac{1}{HW} \sum_{i=0}^{H} \sum_{j=0}^{W} \hat{D}'_k(i,j)$ is the global average pooling over spatial units of the predicted density map $\hat{D}'_k$ corresponding to image $I'_k$ in the pair, $\hat{p}_k = \frac{1}{HW} \sum_{i=0}^{H} \sum_{j=0}^{W} \hat{D}_k(i,j)$ the global average pooling of $\hat{D}_k$ corresponding to $I_k$, and $\epsilon$ is a ranking margin set to zero here. It is known from the cropping and ordering within pairs, that $p' \leq p$ and thus if the model predicts this order correctly the loss value for this pair will be 0. Otherwise the difference of the two will be added to the total loss: the greater the difference is, the greater the increase in loss. This way the model can learn the correct order within a pair according to number of fish [64]. It is not necessary to know the exact count of either image, hence enabling the self-supervised task.

We trained this task together with the supervised one in a multi-task network. The model is end-to-end trained, simultaneously learning to rank the unlabelled data and count the number

of fish in labelled data. The weights are shared across all branches, which means no additional layers are included and the number of trainable parameters is the same as for the supervised on its own. The input to the multi-task model is a mini-batch consisting of $K$ images from the labelled dataset and $K$ pairs of images from the unlabelled dataset. The ranking loss is simply added to the supervised counting loss $\mathcal{L}_{\text{cau}}$ to achieve this:

$$\mathcal{L} = \mathcal{L}_{\text{cau}} + \mathcal{L}_{\text{r}} \tag{6}$$

As shown later in the experiments, training the self-supervised ranking task (with more training data) improves the performance and generalisability of the supervised task of counting fish in labelled image data. We therefore know that any improvement in results is not due to more complexity in the model.

## Results

### Experimental setup

**Dataset split and evaluation metrics.** The labelled dataset was randomly split into a hold-out partition of 350 training images, 70 validation, and 80 test. We made sure that the distribution of data in these sets was reasonably consistent to minimise bias in results. Validation is used for early-stopping during training and hyperparameter optimisation. Once trained, we ran inference on the test data to analyse performance.

Following the common practice in counting by deep learning literature, we evaluate test results using the Mean Absolute Error (MAE) and Root Mean Squared Error (RMSE):

$$MAE = \frac{1}{N_{L_{\text{test}}}} \sum_{k=1}^{N_{L_{\text{test}}}} |c_k - \hat{c}_k| \tag{7}$$

$$RMSE = \sqrt{\frac{1}{N_{L_{\text{test}}}} \sum_{k=1}^{N_{L_{\text{test}}}} (c_k - \hat{c}_k)^2} \tag{8}$$

where $N_{L_{\text{test}}}$ is the number of test samples.

**Information-entropy-based balance regularization.** For further comparison, we also validate our uni-task and multi-task networks with and without a balance regularizer [51] in a study that is, to the best of our knowledge, the only other attempt to count fish in multi-beam sonar images with deep learning. In that study [51], the authors incorporated a regularizing technique to increase the weight of less common samples in their dataset, i.e. those with high numbers of fish. This in turn increased the overall accuracy of the network's predictions, particularly for this subgroup of samples. Our dataset is imbalanced, as commonly seen with data collected in the wild [51]. There are a far greater number of images containing less than 50 fish compared to those with several hundred. We therefore adjust our loss function, which weights samples during online compilation of mini-batches. We group all images into 3 classes according to number of fish, $c$:

- Class 1: 75% of images, $c < 50$

- Class 2: 18% of images, $50 \leq c < 150$

- Class 3: 7% of images, $c \geq 150$

The following "information-entropy-based" balance regularizer (IEB-reg), $\mathcal{L}_{\text{ieb}}$, is then applied so that the weight given to a sample is negatively correlated with the number of

samples of its same class in the batch. For a batch of $K \leq N_L$ images:

$$\mathcal{L}_{\text{ieb}} = -\sum_{k=1}^{K} \log\left(\frac{K_{\text{class}(k)}}{K}\right)|c_k - \hat{c}_k| \tag{9}$$

where $K_{\text{class}(k)}$ is the number of images of the same class as $I_k$ in the batch. The $\mathcal{L}_{\text{ieb}}$ regularizing term is then added to the absolute loss function $\mathcal{L}_c$:

$$\mathcal{L} = \mathcal{L}_c + \lambda\mathcal{L}_{\text{ieb}} \tag{10}$$

We found the hyperparameter λ optimal at 0.1 for our study. Higher values caused the term $\mathcal{L}_{\text{ieb}}$ to dominate over $\mathcal{L}_c$ and meant the model was unable to properly learn. Note this is different from [51] who used λ = 1 but they worked with patches of up to 8 fish meaning much smaller absolute differences in dense inputs. In contrast, the larger absolute differences in our patches can be over 50x greater than the smaller absolute differences. This high absolute difference is also seen in less common images which the regularizer adds more weight to.

## Ablation experiments

To evaluate the performance of our overall methodology and the different individual regularizing components, we carried out 9 ablation studies (Table 1). All 9 trained models were tested on the 80-image test set for comparison. In order to alleviate the stochastic behavior caused by the random initialization of the deep models' weights and the optimization algorithm, we run 3 trials of each ablation study and average across these.

Out of the 9 ablation studies, the method that combined all our proposed regularizing techniques, the multi-task with aleatoric uncertainty regularization, MT + AU-reg (viii), achieved the lowest MAE score of 6.48. The pure multi-task network, MT (vi), achieved the lowest RMSE score 14.27. These are a marked improvement from the outcome of the baseline uni-task network, UT (i), where MAE and RMSE of predictions was 11.09 and 23.88 respectively and the models where the balance regularizer (IEB-reg) [51] was implemented, UT + IEB-reg (ii) and MT + IEB-reg (vii).

Results of the experiments are shown in Table 2. Additionally, Fig 5(a) shows the results of each method split by 5 subgroups of the data according to the number of fish present to understand the effect of each ablation study on them. The subgroups shown are in line with the

**Table 1. Overview of 9 ablation studies.**

| Method | Loss function | Number of train samples | Weight initialization |
|---|---|---|---|
| (i) Uni-task (UT) | $\mathcal{L} = \mathcal{L}_c$ | 350 | ImageNet // Xavier |
| (ii) + IEB-reg | $\mathcal{L} = \mathcal{L}_c + \lambda\mathcal{L}_{\text{ieb}}$ | 350 | ImageNet // Xavier |
| (iii) + AU-reg | $\mathcal{L} = \mathcal{L}_{\text{cau}}$ | 350 | ImageNet // Xavier |
| (iv) + IEB-reg & AU-reg | $\mathcal{L} = \mathcal{L}_{\text{cau}} + \lambda\mathcal{L}_{\text{ieb}}$ | 350 | ImageNet // Xavier |
| (v) + augmented data | $\mathcal{L} = \mathcal{L}_c$ | 5,672 | *UT (i)* |
| (vi) Multi-task (MT) | $\mathcal{L} = \mathcal{L}_c + \mathcal{L}_r$ | 5,672 + 5,672 pairs | *UT (i)* |
| (vii) + IEB-reg | $\mathcal{L} = \mathcal{L}_c + \mathcal{L}_r + \lambda\mathcal{L}_{\text{ieb}}$ | 5,672 + 5,672 pairs | *UT + IEB-reg (ii)* |
| (viii) + AU-reg | $\mathcal{L} = \mathcal{L}_{\text{cau}} + \mathcal{L}_r$ | 5,672 + 5,672 pairs | *UT + AU-reg (iii)* |
| (ix) + IEB-reg & AU-reg | $\mathcal{L} = \mathcal{L}_{\text{cau}} + \mathcal{L}_r + \lambda\mathcal{L}_{\text{ieb}}$ | 5,672 + 5,672 pairs | *UT + AU-reg (iii)* |

Uni-task (UT) models (i-iv) were initialised with ImageNet weights [63, 68] (ResNet-50 backbone) and Xavier initialisation method [69] (final convoluational layer). UT + augmented data (v) and the multi-task (MT) models (vi-ix) were initialised with trained weights from their comparative UT network. To maximise chances of success, MT + IEB-reg & AU-reg (ix) was initialised with UT + AU-reg (iii) instead of UT + IEB-reg & AU-reg (iv), as (iii) produced better results than (iv).

categories chosen for assigning weights according to the balance regularizer (IEB-reg). Except the most common group of $c < 50$ fish / sample has been broken down further to show results for samples with less than 25 fish separately. The reason being that these images will be mostly sparsely distributed and relatively easy to count. Beyond 25 fish, more occlusions between individuals will occur. Samples which contain large elements of noise, usually either dolphins or fishing nets, are also shown as a separate subgroup to examine how each model, with associated loss term, handles this particular challenge. Instead of plotting the MAE, the absolute error in a given sample prediction has been divided by the average ground truth fish count for that group ("Normalised MAE" (NMAE), y-axis). The MAE across a subgroup $g$ has thus been somewhat normalised and so can more reasonably be compared between subgroups:

$$\text{NMAE}(g) = \frac{\sum_{k=1}^{M_g} |c_k - \hat{c}_k|}{\sum_{k=1}^{M_g} c_k} \tag{11}$$

where $M_g$ is the number of samples within a subgroup $g$.

**Ablation on aleatoric uncertainty regularization.** When modifying the network architecture and loss function with aleatoric uncertainty regularization (AU-reg), to quantify, alongside the prediction for fish abundance, the uncertainty in the prediction, $\hat{\sigma}_k^2$, UT + AU-reg (iii) achieved more accurate predictions than the baseline model UT (i). MAE decreased from 11.09 to 8.89, a 20% reduction, and RMSE decreased from 23.88 to 20.24, by 15% (Table 2).

**Ablation on data augmentation.** UT + augmented data (v) which was trained on the same UT (i) architecture and loss function, but where training data has been augmented to increase the number of samples from 350 to 5,672, improves the baseline UT (i) MAE score from 11.09 to 7.88 and RMSE score from 23.88 to 17.20 (Table 2). This equates to a 28% reduction in both error scores.

**Ablation on self-supervised task with unlabelled data (multi-task network).** As the UT + augmented data (v) showed that training with additional synthetic data, notably improved performance, we train all our multi-task networks (vi-ix) with this larger labelled dataset and compare our results with (v). MT (vi) which adds the pair-wise ranking hinge loss to the loss term and trains on unlabelled data as well as the labelled samples, reduces the MAE score further from 7.88 to 7.05 (11% decrease), and RMSE from 17.20 to 14.27 (17% decrease) from UT + augmented data (v) (Table 2). It actually achieved the lowest RMSE score out of all approaches taken, showing it is the least susceptible to extreme values. Fig 5(a) allows for a greater understanding of what is driving this improvement: the last two columns show MT

**Table 2. MAE and RMSE results of 9 ablation studies.**

| Method | Average | | Experiment 1 | | Experiment 2 | | Experiment 3 | |
|---|---|---|---|---|---|---|---|---|
| | MAE | RMSE | MAE | RMSE | MAE | RMSE | MAE | RMSE |
| (i) Unit-task (UT) | 11.09 | 23.88 | 10.65 | 22.73 | 11.88 | 25.44 | 10.74 | 23.47 |
| (ii)    + IEB-reg | 10.27 | 22.01 | 10.28 | 21.91 | 11.52 | 26.99 | 9.00 | 17.13 |
| (iii)    + AU-reg | 8.89 | 20.24 | 8.79 | 19.16 | 8.37 | 20.45 | 9.50 | 21.10 |
| (iv)    + IEB-reg & AU-reg | 11.27 | 25.22 | 11.09 | 23.03 | 11.02 | 25.20 | 11.69 | 27.42 |
| (v)    + augmented data | 7.88 | 17.20 | 7.94 | 16.48 | 8.17 | 17.35 | 7.53 | 17.78 |
| (vi) Multi-task (MT) | 7.05 | **14.27** | 7.18 | 15.39 | 6.54 | 13.88 | 7.42 | 13.53 |
| (vii)    + IEB-reg | 7.87 | 16.67 | 9.91 | 21.81 | 6.16 | 13.13 | 7.53 | 15.07 |
| (viii)    + AU-reg | **6.48** | 14.81 | 6.26 | 13.66 | 6.31 | 15.12 | 6.88 | 15.65 |
| (ix)    + IEB-reg & AU-reg | 7.25 | 16.99 | 7.64 | 19.26 | 6.44 | 16.02 | 7.67 | 15.67 |

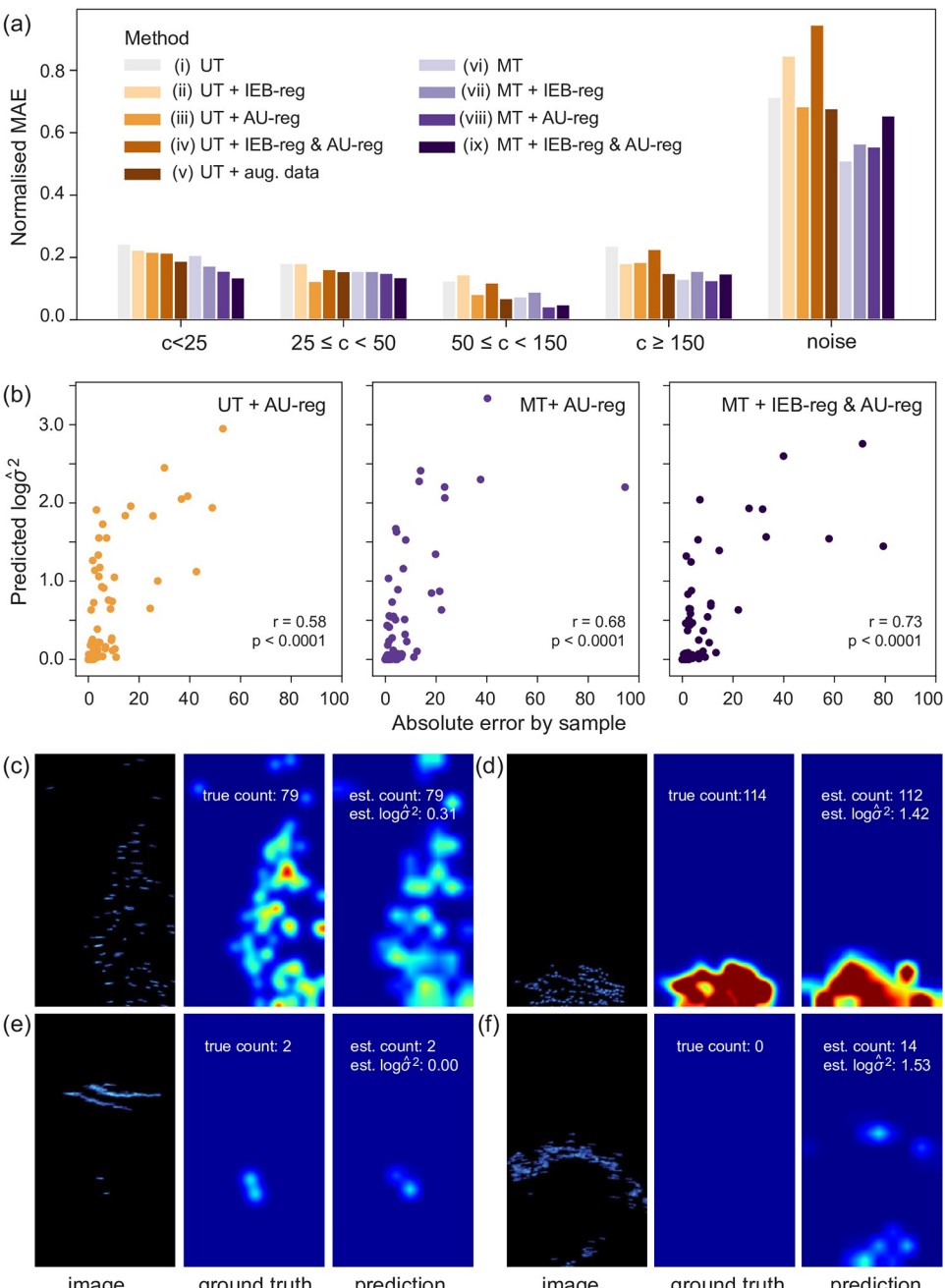

**Fig 5.** Performance of the deep learning models for counting fish in sonar images: (a) Error analysis for sample subgroups—categorised by number of fish or noise present. The MAE for a sample has been divided by the average actual count within each subgroup so results are somewhat normalised and can be compared between subgroups (Eq 11). The percentage of samples that fall within each group are: $c < 25$: 34%, $25 \leq c < 50$: 10%, $50 \leq c < 150$: 14%, $c \geq 150$: 9%, Noise: 34%. The reason $c < 50$ (our first class for balance regularization) is altogether lower than 75% is because many of these samples have been put into the "noise" subgroup for this analysis. (b) The relationship between predicted noise variance and absolute error score, for models with AU-reg (iii, viii, ix). (c-f) Four sample images with corresponding ground truth and predicted density map. Predicted density maps from our best performing model, MT + AU-reg (viii). The density maps can be interpreted as a typical heat map where areas of red indicate dense regions of mullet.

(vi) better predicts samples with high numbers of fish and samples with noise compared with (v).

**Ablation on multi-task network with aleatoric uncertainty regularization.** When combining all three techniques in a multi-task network with aleatoric uncertainty regularization (and training on the larger labelled dataset with synthetic images), MT + AU-reg (viii), we achieved the lowest MAE error score out of all the approaches tested. MAE was 6.48, a 0.57, or 8%, reduction on the second best score from MT (vi). This is a 4.61, or 42% reduction in MAE from baseline UT (i), where none of these techniques were implemented (Table 2). MT + AU-reg (viii) achieves the best results (lowest NMAE score on the bar plot) for more densely populated fish images with limited noise, where images contain more than 50 fish (Fig 5(a), columns 3 and 4). It achieved the second best error score, behind MT (vi), for images containing substantial noise (Fig 5(a), last column).

With uncertainty regularization, we also obtain a measure of prediction uncertainty $\hat{\sigma}^2$, alongside the prediction of fish abundance. In Fig 5(b), the measure of prediction uncertainty (outputted as $\log \hat{\sigma}^2$) is plotted against the absolute error of the count prediction for the best performing three out of four methods where uncertainty regularization has been incorporated: UT + AU-reg (iii), MT + AU-reg (viii) and MT + IEB-reg & AU-reg (ix). There is a moderate positive correlation between the absolute error score and estimated uncertainty of a sample. For the two multi-task networks, the correlation statistic (r) reaches 0.68 and 0.73, without and with the IEB-reg respectively ($p < 0.001$, one-tailed test, in all cases). The noise variance predictions, or uncertainty measure, are heavily skewed towards lower values, seen in the scatter plots where a high number of samples are clustered in the bottom (left) corners. Specifically, MT + AU-reg (viii) gives a prediction uncertainty score as $0 \leq \log \hat{\sigma}^2 < 1.7$ for 90% of samples and $1.7 \leq \log \hat{\sigma}^2 < 3.7$ for just 10% of samples. In fact, nearly 50% of samples have predicted $\log \hat{\sigma}^2 = 0$.

Four test sample images along with their corresponding ground truth and predicted density maps are shown in (Fig 5c–5f) so results can be compared locally. The density maps can be interpreted like typical heat maps, where areas of red indicate dense regions of mullet. Prediction outputs are from MT + AU-reg (viii), the best performing model. In image Fig 5(c), where mullet are in relatively high numbers with some occlusions between fish, we can see the model gives a perfect prediction and a relatively low uncertainty measure of 0.31, so we can be confident in this prediction. This is in contrast to Fig 5(d), where the number of fish is similar but they are seen more densely compact. Differences in the sonar video settings can also cause variability between images, for example anti-aliasing control can introduce distortion in the image as seen here, and so it is more difficult to distinguish between individual mullet. Whilst the model's prediction is within 98% accuracy (error = -2), the uncertainty score is relatively high ($\log \hat{\sigma}^2 = 1.42$) for these reasons. We can see this discrepancy in uncertainty score when different types of noisy objects dominate the image: in Fig 5(e) the model is able to distinguish between dolphin and fish confidently ($\log \hat{\sigma}^2 = 0$) and gives a perfect prediction. But with the fishing net present in Fig 5(f), the uncertainty score is high, 1.53, within the highest 15% of scores. Whilst the model clearly recognises most of the noisy object correctly as not mullet, the blue blobs at the middle bottom of the image have been "incorrectly" predicted as mullet. This is difficult to decipher for even an expert human labeller: these could be mullet as the model predicts or splashes from the net (how they have been annotated). This uncertainty score is meaningful.

## Implementation and training procedure details

The ResNet-50 backbone of the uni-task networks (i-iv), was initialised with ImageNet weights [63, 68]. Transfer learning in this way has been proved effective in related computer vision

research [64]. Any additional layers were initialised with Xavier initialisation method [69]. These models were trained for 300 epochs. To mitigate the effect of randomness in training deep models (due to the descent algorithms needed for optimising many parameters), ablation studies UT + augmented data (v), and the multi-task architectures (vi-ix), were initialised with weights from corresponding uni-task models (i-iv). The incremental effect of the synthetic data and the self-supervised task could then be evaluated. These models were trained for a further 200 epochs. The Adam Optimizer [70] was used for minimising the loss term, with a learning rate of $10^{-4}$. For uni-task models (i-iv), which were initialised from the start (with ImageNet weights, [63, 68]), we lowered the learning rate to $10^{-5}$ after 200 epochs. As for the size of the mini-batches, we set $K = 10$. In the case of the multi-task models, the number of pairs is also $K$, meaning 10 labelled images and 10 pairs of images are used in each training iteration (Fig 4). When generating ground truth density maps, $s = 4$ and standard deviation $\sigma = 1$ were used.

All models were built with Tensorflow 2.2 and Keras API.

## Transfer learning and comparison with state-of-the-art approaches

**Comparison with information-entropy-based balance regularization.** Aleatoric uncertainty regularization, AU-reg, outperformed the IEB-reg for both the uni-task, 8.89 vs 10.27 MAE, and multi-task network, 6.48 vs 7.87. For a thorough evaluation, we also experimented with modifying the loss term to include both the balance and uncertainty regularizer: UT + IEB-reg & AU-reg (iv) and MT + IEB-reg & AU-reg (ix), but performance was worse when comparing to the baselines, UT (i) and MT (iv) and the networks with AU-reg only, UT + AU-reg (iii) and MT + AU-reg (viii) (Table 2).

**Evaluation on DeepFish dataset.** To demonstrate the generalisability of our proposed counting framework, MT + AU-reg (viii), we tested it on the recently published, publicly available benchmark dataset DeepFish [42]. DeepFish contains 40k underwater images from 20 different coastal and nearshore benthic habitats in Australia, with a binary classification label, either "fish" or "no fish". Alongside this, a subset of 3,200 images are annotated with a simple point annotation to mark the number of individual fish (Fig 6). This "counting" subset of data can therefore be used to train and test counting schemes. Data were collected in full HD resolution from a digital SLR underwater camera and there is a mean of 1.2 fish/image, ranging from 0–18 fish. This contrasts to our dataset with much lower visibility and mean of 42 fish/ image, range 0–438.

To implement our framework we incorporated a ranking task alongside the counting task, making use of the larger, weakly labelled, classification dataset: the parallel Siamese

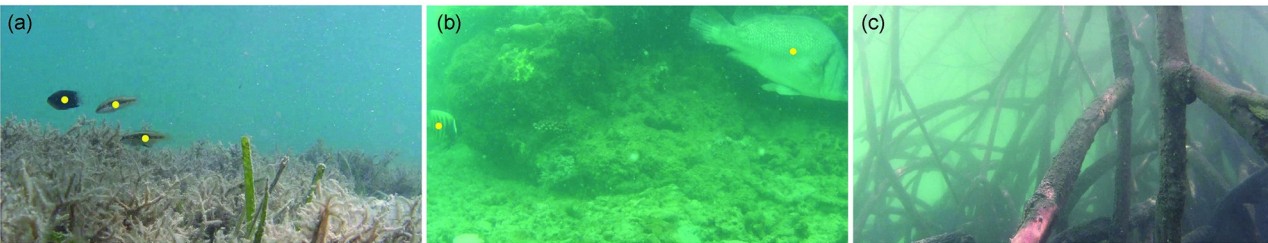

**Fig 6. Three sample images from the "counting" subset of the DeepFish dataset with point level annotation.** All data were collected with HD resolution digital cameras in 20 different marine habitats in tropical Australia. Mean of 1.2 fish/image, ranging from 0–18 individuals. (a) Low algal bed, count: 3, classification: "fish" (b) Reef trench, count: 2, classification: "fish" (c) Upper mangrove, count: 0, classification: "no fish". The images were obtained from the open-source dataset DeepFish [42], licensed under a Creative Commons Attribution 4.0 International License.

**Table 3. MAE result on DeepFish dataset.**

| Method | MAE |
|---|:---:|
| i) DeepFish (benchmark) [42] | 0.38 |
| ii) Ours—Multi-task + AU-reg (without TL) | 0.34 |
| iii) Ours—Multi-task + AU-reg (with TL) | **0.30** |

Comparison between DeepFish authors' benchmark result and our model, with and without transfer learning (TL). "Without TL": initalised with ImageNet weights [63, 68], "with TL": initialised with pre-trained weights from previous training on our novel sonar image dataset.

architecture of our network can again learn to rank pairs of images as it is known that an image labelled as "fish" has a greater number of fish than an image labelled with "no fish". As there were so few fish / image it did not make sense to take random crops of images (Algorithm 1), but we were still able to leverage large quantities of data that otherwise could not be used in the supervised counting branch alone. Compared to the proposed benchmark model [42], we reduced the MAE for fish count by 21%, 0.30 vs 0.38 (Table 3). We also demonstrate that this implementation benefits from transfer learning when initialising the new model with the weights of our previously model trained with our novel dataset: the MAE is lower when we initialised the model with weights from previous training on our dataset versus the ImageNet weights, 0.30 vs 0.34 (Table 3).

## Discussion

Our findings exemplify how deep learning by regression can be effectively employed to count aquatic organisms in underwater images and we present a successful, open-source framework along with novel training data. We leveraged abundant unlabelled data in a self-supervised task, to enhance performance of the direct supervised task, to count mullet fish with high accuracy in challenging low-resolution sonar imagery. By combining this with uncertainty regularization, accuracy not only improved but we provide a meaningful quantification of prediction uncertainty to the user for informed biological decision-making. Our proposed multi-task network with uncertainty regularization, MT + AU-reg (viii), achieved an MAE of 6.48, on images containing up to 438 mullet. This was a 42% improvement in error over our baseline deep learning model where no regularization techniques were implemented. Moreover, we introduce a new large dataset of sonar videos and a fully annotated subset of 500 images [60].

While the demand for processing large volumes of visual data to assess natural populations led a surge in developing computer vision tools, traditional methods are limited in their generalisability. The success of deep learning in broader image analysis is encouraging, but the lack of annotated underwater data is one key factor constraining progress of its use to count aquatic animals. The majority of progress has been made in classifying species or detecting individuals where numbers are low and sparsely distributed (e.g. [36, 37, 39]). We advance this by building a widely applicable framework that tackles the task of counting when abundance can be high and occlusions common from schooling fish. This is particularly challenging with low-resolution images, such as from sonar cameras, and when data are collected in an unconstrained, wild environment, where noise and variation is inevitable. The techniques we adopted address these challenges and improve the accuracy and generalisability of a deep network. Furthermore, we demonstrate an improvement when compared to the alternative, state-of-the-art, deep learning methodologyfor analysing underwater data in the form of both comparable low

resolution sonar images [51], namely the proposed balance regularizer, and high-resolution images from a breadth of wild habitats[42].

To ensure robust assessment of our methodology, we carried out 9 different ablation studies with varying training data, model architectures and loss functions, and trialled 3 times each. Prediction error on our test data improved with each novel technique implemented: from innovative ways to augment the annotated data, to building a multi-task network to simultaneously train the self-supervised task, to finally combining this with uncertainty regularisation. All weights are shared across the supervised and self-supervised task, and adding a channel to accommodate for uncertainty prediction added an immaterial number of parameters. Thus, the improvement in performance is not simply due to a more complex model (that may generalise less well). Note also, this is the result from tests on a biased challenging sample, deliberately chosen to contain a significantly higher proportion of noisy samples and those with dense schools of fish.

### Regularizing: Self-supervised task with unlabelled data (multi-task network)

The improvement in prediction error, both MAE and RMSE, from comparing the results of MT (vi) with the uni-task model, UT + augmented data (v) (both trained on the larger labelled dataset with synthetic images), supports our hypothesis that by increasing the size of the training set through unlabelled data, and training the parallel self-supervised ranking task, improves the model's ability to count fish accurately in unseen data. In fact, MT (vi) achieves the lowest RMSE out of all 9 methods. Its performance is notable when predicting samples containing higher numbers of fish and samples with substantial noise (Fig 5(a)). We speculate that this is because by adding significantly more unlabelled data, we likely increase the number of samples that fall within these two challenging categories, which will disproportionately require more training data. Samples with high numbers of fish also make up the smallest percentage of the labelled training data, adding to the likelihood that the uni-task models (i-v) will not be as well adapted to this category. These samples are also the most time-consuming to annotate. Thus, adding more of such images through unlabelled data allows for an efficient way to access these samples to help increase the accuracy of the corresponding predictions. A common outcome of trained neural networks' predictions, is a regression to the mean of training data. The mean of our training data is 42 fish. Thus the multi-task's ability to predict images of extreme high values compared to the uni-task network, is also a positive sign that it is more robust to not simply regressing to the mean.

### Regularizing: Multi-task network with aleatoric uncertainty

To date, computer vision tools that have been developed for calculating fish abundance provide only an estimation of number or biomass (e.g. [43, 51, 52]), so the user will have no knowledge of the uncertainty around a prediction. By contrast, our proposed model incorporates uncertainty regularization to the multi-task network, MT + AU-reg (viii). As well as providing an indication of the inherent noise present in samples resulting in uncertain predictions, this regularization was found to improve MAE in both the UT (i) and MT (vi) models by 20% and 8% respectively, with MT + AU-reg (viii) achieving the lowest MAE out of all methods. This supports our hypothesis that it improves model training leading to more accurate predictions. Notably, the MT + AU-reg (viii) improves predictions for medium to dense samples of mullet to an even greater extent than MT (vi) (Fig 5(a), columns 3 and 4). Again, this is a positive indication that it is not simply regressing predictions to the mean. It achieved the second lowest error score with samples where substantial noise is present,

comparatively worse than the MT (vi). This is expected because essentially adding this regularizing term, allows the model to ignore noisy images in training so it learns to predict non-noisy images more accurately, but in turn there will be a trade-off in its ability to handle noisy images.

We found that there was a moderate correlation between uncertainty and error in prediction. The uncertainty predictions were also heavily skewed towards a low score (Fig 5(b)). In practice, the user can then choose to treat sample results with high relative noise variance scores with caution or investigate further. These uncertainty predictions are likely to be both few and meaningful in suggesting there could be an error in count prediction, thus manageable and beneficial for further human interpretation. When considering the application of these methods for aquatic monitoring and conservation, or fisheries and aquaculture management, over or underestimating populations could lead to adverse consequences such as biased decisions. Therefore, a greater understanding of how much a prediction can be depended on is crucial.

### Advances relative to previous work on deep learning to count fish

We tested our framework against the regularization method of the only previous study on using deep learning to count schools of fish in sonar images [51]: we incorporated the balance regularizing term therein proposed, IEB-reg, in our uni-task and multi-task networks, UT + IEB-reg (ii) and MT + IEB-reg (vii), and ran these as additional experiments. In both cases, the pure respective models, UT (i) and MT (vi), and these models with uncertainty regularization, UT + AU-reg (iii) and MT + AU-reg (viii), performed better than the models with IEB-reg in terms of overall prediction error (Table 2). We expected the IEB-reg to improve results of the less common, more densely populated images as it increases the weight of these samples within a batch. This was the case when comparing UT + IEB-reg (ii) to UT (i), but surprisingly not when comparing MT + IEB-reg (vii) to MT (vi) (Fig 5(a)). One plausible explanation could be overfitting these types of samples during training by adding too much weight, hence, not performing so well on unseen data.

When the networks were trained with both IEB-reg and AU-reg together, it was apparent that these two regularizers do not complement each other. Prediction accuracy was worse than when only AU-reg was implemented as highlighted in the results for samples containing substantial noise (Fig 5(a), last column): UT + IEB-reg & AU-reg (vi) and MT + IEB & AU-reg (ix) produced the highest NMAE (Eq 11) scores out of all the uni-task and multi-task networks, respectively. This likely results from both regularizers reducing the weighting of noisy images (which tend to contain fewer fish) in training and thus when used together, the performance on this type of data in testing is worse.

We also tested our proposed model architecture on a recently published dataset, DeepFish, where visual images were collected with a digital SLR underwater camera and so are of an entirely different format to sonar images. By improving upon the results of the author's benchmark deep model [42], we demonstrate our model can provide reasonably accurate predictions on wide-ranging underwater data: on low-resolution, monochromatic and densely populated images, as well as on high-resolution, sparsely populated images, of varying habitats and species of fish.

Other recent work deployed a hybrid CNN based on a multi-column and dilated CNN to count farmed Atlantic salmon fish with a density-based regression methodology [52]. Here natural images were used and so fish present more distinct features. Data were also collected in an enclosed mariculture net cage so it is unlikely other noisy objects were present and is thus difficult to compare our study directly. A multi-column network is computationally expensive

to train but it would be interesting to incorporate a dilated backend in our network, as in [66], to see if performance can be improved still. The challenge of distinct lack of available labelled datasets was addressed by using a variety of techniques to augment thousands of *side scan* sonar images from a small starting dataset to count fish and dolphins, up to 34 and 3 respectively per image [67]. Different from the Adaptive Resolution Imaging Sonar used in our study, side scan sonars only image non-moving targets which, combined with much fewer fisher detected per image, complicates any direct comparison with our results. Nevertheless, our study supports their findings that it is possible to solve a problem like this even with a relatively small starting labelled dataset by adopting data augmentation techniques. From this, we expand the training data with synthetic data to improve model predictions on unseen sonar data— a widely proven and adopted technique in deep learning research [71]. We show that leveraging unlabelled data and incorporating uncertainty regularisation, together with data augmentation, improves performance further.

## From technical to applied relevance

Beyond the technical merit and relevance in solving the task of processing low-resolution sonar-based underwater footage for the broader application of quantifying abundance of other underwater species, there is also a practical and conservation relevance in this context in the ability to automatically and accurately count mullet fish at the spatial scale that matters for the dolphin-fisher interaction in southern Brazil. To properly evaluate the speculative benefits of this interaction, the first steps are quantifying precisely both the (1) availability of mullet schools at the very local scale at which the dolphin-fisher interaction takes place; and (2) the proportion of the mullet available that is caught by fishers and dolphins when interacting and when foraging independently. Quantifying these benefits for interacting dolphins and fishers is crucial to determine whether their interactions are indeed mutual and, if so, to determine the minimum conditions of prey availability at which this traditional, century-old interaction can persist and remain resilient in face of the global trend of decline fisheries seen at local, regional and global scales (e.g. [3–5]). Given the real concern that the mullet stocks in southern Brazil are in decline [72] and causing a decline in the frequency of dolphin-fisher interactions, evaluating whether these changes can collapse this unique socio-ecological system becomes imperative. Our deep learning tool can now be used to efficiently and effectively process the more than 1 million images collected in field sampling and used to infer an estimation of the local mullet population. Corresponding uncertainty predictions alongside each sample count prediction, will allow for a greater understanding in how much each automated image count can be trusted, the first time this measure has been incorporated for counting fish with deep learning.

From labelling a dataset such as this, we understand the challenges it presents. It is difficult even for the biological experts to determine fish numbers in very dense images and at times distinguish between noise and fish. To this end, there will very likely be inaccuracies, bias, and even inconsistencies in the labelling which will have affected the training capacity of the model and lead to discrepancies between predictions and ground truths. This is likely the cause of many errors in predictions—i.e. not necessarily the limitations of such a model in its capability of image analysis. Expanding this dataset with more labelled images and using multiple annotators to reduce noise by consensus after removing bad annotations based on inter-rater agreement metrics [73, 74] would be a beneficial development, as well as training the deep network on data from wider biological research and introducing the temporal dimension to capture motion information.

## Conclusion

Effective tools for monitoring fish stocks in the wild are required to support conservation efforts worldwide as well as wide-ranging biological research [9, 10]. Imaging devices and computer vision systems can be hugely valuable in being able to automatically access underwater populations in a cheap, efficient, and non-intrusive way. However, automatically counting fish in video and image data is a challenging task, particularly when dense schools of fish and/ or substantial noise is present. While traditional computer vision methods are limited in their generalisability, deep learning for counting aquatic animals in video and imagery is largely unexplored, especially in comparison to research carried out in other applications. One of the potential reasons for this is the lack of large publicly available underwater video/image datasets and the difficulty and cost of having them manually and accurately annotated to train powerful deep models. In this context, we provide a novel dataset of sonar video footage of mullet fish recorded in a specific, special, cooperative foraging system between bottlenose dolphins and artisanal fishers [60]. From this large dataset, 500 images were manually labeled with point annotations locating the contained fish. Moreover, we developed an effective density-based deep learning approach to automate the process of quantifying fish from sonar images. Our multi-task network learnt to count fish by regressing an image to its representational density map, exploiting the labelled data, while learning to rank pairs of unlabelled images by their relative fish count as a way to leverage useful information for the counting task and make the most of unlabelled data. To favour interpretability and further improve the results, aleatoric uncertainty was estimated alongside the density maps by our network. The results obtained demonstrate the effectiveness of our techniques and, in particular, the possibility of using them to analyse a wider range of underwater imaging as, for instance, with traditional underwater photography, such as the DeepFish dataset. In providing an open-source framework for practical use, our study puts forth a template for crowd counting animals. We make our data, code and pre-trained network available to benefit the advancement of other counting models via transfer learning, thereby contributing to the continuous development of methods for assessing natural populations in times of biological crisis.

## Supporting information

**S1 Video. Sonar video sample.** Underwater video generated by an Adaptive Resolution Imaging Sonar (ARIS 3000, Sound Metrics Corp, WA, USA) displaying the top view of a passing mullet school (*Mugil liza*) before a fishing net is cast by artisanal fishers in Laguna, southern Brazil. The video was manually pre-processed for contrast enhancement and background removal.
(MP4)

## Acknowledgments

We are grateful to FG Daura-Jorge and DR Farine for the field logistics and support during fieldwork, to all researchers involved in data sampling (Machado AMS, Bezamat, B Romeu, JVS Valle-Pereira, PV Castilho, BS Silva, N da Silva, LF da Rosa, CF Alves, L Conti, D Klein, M Rodrigues), and to the anonymous referees for their insightful comments on the manuscript.

## Author Contributions

**Conceptualization:** Penny Tarling, Mauricio Cantor, Sergio Escalera.

**Data curation:** Penny Tarling, Mauricio Cantor.

**Formal analysis:** Penny Tarling, Albert Clapés, Sergio Escalera.

**Funding acquisition:** Mauricio Cantor, Sergio Escalera.

**Investigation:** Penny Tarling.

**Methodology:** Penny Tarling, Mauricio Cantor, Albert Clapés, Sergio Escalera.

**Project administration:** Sergio Escalera.

**Resources:** Mauricio Cantor, Albert Clapés, Sergio Escalera.

**Software:** Penny Tarling, Albert Clapés.

**Supervision:** Mauricio Cantor, Albert Clapés, Sergio Escalera.

**Validation:** Penny Tarling, Albert Clapés, Sergio Escalera.

**Visualization:** Penny Tarling, Mauricio Cantor, Albert Clapés.

**Writing – original draft:** Penny Tarling, Mauricio Cantor.

**Writing – review & editing:** Penny Tarling, Mauricio Cantor, Albert Clapés, Sergio Escalera.

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
