## [Decision Letter · Decision Letter 0]

10 Jan 2022

PONE-D-21-23858Deep learning with self-supervision and uncertainty regularization to count fish in underwater imagesPLOS ONE

Dear Dr. Cantor,

Thank you for submitting your manuscript to PLOS ONE. After careful consideration, we feel that it has merit but does not fully meet PLOS ONE’s publication criteria as it currently stands. Therefore, we invite you to submit a revised version of the manuscript that addresses the points raised during the review process. 

We look forward to receiving your revised manuscript.

Kind regards,

Tim Wilhelm Nattkemper

Academic Editor

PLOS ONE

Journal Requirements:

2. We note that Figures 1 and 6 in your submission contain copyrighted images. All PLOS content is published under the Creative Commons Attribution License (CC BY 4.0), which means that the manuscript, images, and Supporting Information files will be freely available online, and any third party is permitted to access, download, copy, distribute, and use these materials in any way, even commercially, with proper attribution. For more information, see our copyright guidelines: http://journals.plos.org/plosone/s/licenses-and-copyright.

a. You may seek permission from the original copyright holder of Figures 1 and 6 to publish the content specifically under the CC BY 4.0 license. 

Additional Editor Comments:

Dear authors, finally, the reviews have been completed and both reviewers agree that minor revisions are necessary before your submission can be accepted. Please pay attention to the comments (one reviewer submitted an extra file with comments, hope this will be included / attached by the PLoS system) and follow them carefully.

Reviewers' comments:

Reviewer's Responses to Questions

**Comments to the Author**

1. Is the manuscript technically sound, and do the data support the conclusions?

Reviewer #1: Yes

Reviewer #2: Yes

2. Has the statistical analysis been performed appropriately and rigorously? 

Reviewer #1: Yes

Reviewer #2: Yes

3. Have the authors made all data underlying the findings in their manuscript fully available?

Reviewer #1: Yes

Reviewer #2: Yes

4. Is the manuscript presented in an intelligible fashion and written in standard English?

Reviewer #1: Yes

Reviewer #2: Yes

5. Review Comments to the Author

Reviewer #1: 1. The manuscript is technically sound and the data supports the conclusions. Some minor adjustments are needed to clarify some sections (See attachment).

2. The statistical analysis performed is associated with proven machine vision processes.

3. Data is available.

4. Manuscript is well-written.

Reviewer #2: Monitoring underwater populations is crucial to support the conservation of wild aquatic biota. The present manuscript proposes a deep learning method to estimate the number of fish in underwater images.

The authors use a novel dataset of (low-resolution and site-specific) sonar images to train and test various density-based deep learning models. Finally, they argue the generalisability of their approach by testing the best performing model (the one with self-supervision and uncertainty regularization) on a contrasting dataset (high-resolution and heterogeneous-habitat) of underwater photos.

This is a really interesting contribution that fits well with the scope of Plos One, but I have a few minor issues concerning the analysis and some specific choices made by the authors. The manuscript might be considered acceptable after minor revision.

Main comments:

1) Referring to Equation 1, it is unclear to me how the count is extracted out of the predicted density map. Furthermore, I was not able to gain insights on this matter looking at the references [41-45,60]. From what I have understood, a pixel of the predicted density map contains a real number; how is the image predicted density normalized? Does the sum over spatial units provide a real number or an integer one (as it is reported in the estimated count in Figure 5 (c)-(f))?

2) In subsection 'Ablation experiments', citing "All 9 trained models were tested on the 80-image test set for comparison and averaged across the 3 trials". The 3 trials are never mentioned before. I guess they refer probably to different simulation's run, each one with a diverse random seed. Maybe the authors can explicitly state it in the text. In this regard, do the results shown in Figure 5 (a), (b) refer to a specific trial or to an average of the 3?

3) The authors did not provide a reason for the choice of the hyperparameters used for training. Have they performed an hyperparameter optimization? Is the loss function of the models converging? Showing the learning curves (e.g. in Supplementary Materials) may provide useful informations.

6. PLOS authors have the option to publish the peer review history of their article (what does this mean?). If published, this will include your full peer review and any attached files.

Reviewer #1: No

Reviewer #2: No

---

## [Author Response · Author response to Decision Letter 0]

14 Feb 2022

Ref.: PONE-D-21-23858

Title: Deep learning with self-supervision and uncertainty regularization to count fish in underwater images

Authors: Penny Tarling, Mauricio Cantor, Albert Clapés, Sergio Escalera

Authors’ replies to the comments by the Editor

Comment#1, Editor. Dear authors, finally, the reviews have been completed and both reviewers agree that minor revisions are necessary before your submission can be accepted. Please pay attention to the comments (one reviewer submitted an extra file with comments, hope this will be included / attached by the PLoS system) and follow them carefully.

Authors’ reply: Thank you for the very positive evaluation of our work. We are pleased to know that both reviewers welcomed our manuscript and requested only minor revisions. Please see below how we addressed each of them, and paid close attention to all the formatting requirements.

Comment#2, Editor. Please ensure that your manuscript meets PLOS ONE's style requirements, including those for file naming. The PLOS ONE style templates can be found at 

Authors’ reply: Thank you for the attentive review. We have used the PLOS ONE LaTeX templates and double-checked all the formatting requirements.

Comment#3, Editor. 2. We note that Figures 1 and 6 in your submission contain copyrighted images. All PLOS content is published under the Creative Commons Attribution License (CC BY 4.0), which means that the manuscript, images, and Supporting Information files will be freely available online, and any third party is permitted to access, download, copy, distribute, and use these materials in any way, even commercially, with proper attribution. For more information, see our copyright guidelines: http://journals.plos.org/plosone/s/licenses-and-copyright. 

a. You may seek permission from the original copyright holder of Figures 1 and 6 to publish the content specifically under the CC BY 4.0 license. 

b. If you are unable to obtain permission from the original copyright holder to publish these figures under the CC BY 4.0 license or if the copyright holder’s requirements are incompatible with the CC BY 4.0 license, please either i) remove the figure or ii) supply a replacement figure that complies with the CC BY 4.0 license. Please check copyright information on all replacement figures and update the figure caption with source information. 

If applicable, please specify in the figure caption text when a figure is similar but not identical to the original image and is therefore for illustrative purposes only.

Authors’ reply: We appreciate your guidance for re-using published figures. For Figure 1, we clarify that only part (a) is copyrighted by SoundMetrics Coorp, and that parts (b) and (c) are properties of the authors of this manuscript. Note that in Figure 1(a) we present a similar, modified version of their original figure, after having contacted the copyright owner. For full transparency, we have now obtained their written permission, uploaded the Content Permission Form, and updated the Figure 1 caption to clarify that part (a) is similar but not identical to the original image and is therefore for illustrative purposes (page 3).

As for Figure 6, we clarify that the original images were published under the Creative Commons Attribution License (CC BY 4.0), allowing us to reproduce them here since we have cited the source fully and accurately (see our original reference [38] Saleh A, Laradji IH, Konovalov DA, Bradley M, Vazquez D, Sheaves M. A realistic fish-habitat dataset to evaluate algorithms for underwater visual analysis. Scientific Reports. 2020;10(1):1–10). Please see a copy of the license and permission to reuse these images here: 

https://s100.copyright.com/AppDispatchServlet?title=A%20realistic%20fish-habitat%20dataset%20to%20evaluate%20algorithms%20for%20underwater%20visual%20analysis&author=Alzayat%20Saleh%20et%20al&contentID=10.1038%2Fs41598-020-71639-x&copyright=The%20Author%28s%29&publication=2045-2322&publicationDate=2020-09-04&publisherName=SpringerNature&orderBeanReset=true&oa=CC%20BY. For full transparency, we rephrased the Figure 6 caption to declare that the original images were published under CC BY 4.0 (page 17).

Comment#4, Editor. Please review your reference list to ensure that it is complete and correct. If you have cited papers that have been retracted, please include the rationale for doing so in the manuscript text, or remove these references and replace them with relevant current references. Any changes to the reference list should be mentioned in the rebuttal letter that accompanies your revised manuscript. If you need to cite a retracted article, indicate the article’s retracted status in the References list and also include a citation and full reference for the retraction notice.

Authors’ reply: We have double-checked all citations and references, and used their correct and latest BibTeX entries.

\f

Authors’ replies to the comments by Reviewer#1

Comment#1, Reviewer#1. The manuscript is technically sound and the data supports the conclusions. Some minor adjustments are needed to clarify some sections (See attachment).

2. The statistical analysis performed is associated with proven machine vision processes.

3. Data is available.

4. Manuscript is well-written.

Authors’ reply: We very much appreciate the positive feedback on our manuscript and the thoughtful comments that helped us clarify important details. Please see below how we addressed each comment -note that line numbers refer to the revised version with in-line tracked changes.

Comment#2, Reviewer#1. Page 2 “Underwater videos and images offer a non-intrusive…” This is not true, several studies have shown that cameras can attract or scare some species of fish, that can be consider an intrusion in the sense that it disrupts the behavior and natural dynamics of fish populations. No method is non-intrusive, or non-biased. Check refs: Bacheler, Comparing relative abundance, lengths, and habitat of temperate reef fishes using simultaneous underwater visual census, video, and trap sampling, 2017; Campbell, Comparison of relative abundance indices calculated from two methods of generating video count data, 2015, for an idea on the biases of several methods, including cameras.

Authors’ reply: Thank you for improving the accuracy of our text. We agree with the reviewer that studies using underwater imaging can disrupt fish behaviour and are inherently imperfect in estimating population parameters, such as abundance. We have rephrased the excerpt to downplay our original overstatement of the non-intrusiveness of underwater videos (page 2), and point the readers to the references provided for a more in-depth comparison of the pros and cons of different sampling methods.

Comment#3, Reviewer#1. Page 3 “only semi-automated commercial tools to process sonar images are available”. This is not entirely true. Softwares like Echoview or ESP3 can process sonar images in bulk, and they also can count fish in them. Echoview has the option to add user developed scripts in case several files from different sources are needed (so it would be fully-automated). I agree with the assertion that there are not many products, readily available, that process sonar images in a fully-automated way, but that depends on the definition of fully-automated. A well-written script in Echoview or ESP3 can do that. I could not find a full copy of [10] to corroborate that they had a similar conclusion. I don't think the advantage of a computer vision model is that it can automatically process vast numbers of images, but that it takes them and processes them as a whole.

Authors’ reply: We appreciate the reviewer’s perspective on these tools that can process sonar images automatically or semi-automatically and agree that the advantage of computer vision models is the ability to process vast number of images as a whole, i.e. speed. Our original argument was based on our best knowledge at the time of writing that more automated tools are not readily available and are limited in scope. For instance, we cited Ref [10] Lankowicz et al. 2020 as a recent example of fishery researchers using the same ARIS sonar camera as us, but needing to manually count fish in each image and therefore they express the need for faster or more automated methods. We quote, “recognize that sonar imaging surveys have limitations as well; data processing alone is a huge challenge due to the sheer number of images that must be processed. […]. However, advances in computer science fields such as machine learning are quickly improving the performance of automated image processing procedures, and thus it will soon be possible for sonar image data to be processed on a similar time scale to net-generated catch data.” While we knew about Echoview, we were unaware of ESP3 - thank you for bringing our attention to it, which we now cite in the revised manuscript. We still believe these tools have their own limitations: ESP3 seems only compatible with "SIMRAD (.raw) and a small number of other formats from single-beam or split-beam sonar cameras”, and whilst it can detect schools of fish, sea bottom etc. it may have limited capability for quantifying numbers of fish. From our understanding of Echoview, depending on the studied system, it may not be as straightforward to semi-automate fish counting; as the Reviewer says, without a custom well-written script. To clearly acknowledge the existence of these tools, we now cite both in our revised manuscript (page 3), and amended this excerpt to highlight the contribution of our proposed computer vision method: the increased speed in processing vast number of images as a whole (page 3).

Comment#4, Reviewer#1. Page 5 “500 were selected for manual labelling”. Was there any criterium for the selection of this amount of images?

Authors’ reply: Thank you for your question - we have updated the manuscript (page 5) to make this clearer. We chose and stratified 500 images to ensure the dataset covered the visual variability with several images in each high level bucket: the samples chosen contained a wide ranging number of fish (between 0 to 438) and we made sure to include substantial noise. To capture any potential variation between the days and time of day (e.g. in water visibility, noise and fish), the images were selected from 48 different videos so there were at least 2 videos from each of the 18 days of sampling, one from morning and another from afternoon when possible (sometimes the weather conditions did not allow us to deploy the camera for the entire day). Within each morning and afternoon, the videos were selected at random. 

Comment#5, Reviewer#1. Page 8. “Incorporating this measure of uncertainty should not only lead to a higher level of accuracy in count predictions through optimising model training”. Is this a hypotesis? 

Authors’ reply: Thank you for the comment. Yes, this is a hypothesis. We have made our manuscript clearer in stating this sentence as a hypothesis (page 8), and discussing our corresponding findings (pages 18-19).

Comment#6, Reviewer#1. Page 8. “Depth of swimming will cause differences in scale but it does not result in great variations here.” Do you have preliminary data to support this?

Authors’ reply: We do not have preliminary data on this, but we contacted the manufacturer of the ARIS sonar camera to better understand how it operates and the extent to which the acoustic beams could distort the fish images within the spatial scale of our sampling. We realized that the influence of water depth on measuring large fish at short distances with the ARIS would be negligible. The average body length, from the tip of the mouth to the caudal fin, of mullets caught and measured manually during our fieldwork was 42.9 cm ± 7.00 SD (n=771). Such measurable size should be very consistent within the sonar range, as long as the ARIS sonar camera is tilted down enough for the energy beams to “illuminate” the fish and minimize any potential interference from returns caused by the water surface reflections superimposing the fish images. We have rephrased this except to clarify that the way we sample these images in short ranges and in shallow waters should not distort the size of the mullets (page 9). For reference, we also amended the caption of Fig. 1c to give a sense of how large the mullet fish are (page 3).

Comment#7, Reviewer#1. Page 9. “we can leverage some potentially useful information for the original counting task”. What kind of information?

Authors’ reply: We are referring to the comparative number of fish between two images learnt in the self-supervised task. Thank you for pointing this out - we have updated this sentence to clarify our point (pages 9-10).

Comment#8, Reviewer#1. Page 14. “the model is able to distinguish between dolphin and fish confidently”. This classification may not be significant since the size of a dolphin is much greater than the one of a mullet so it is easily differentiated. The uncertainty introduced by the nets is more significant since the size of the "blobs" are comparable to the size of the mullets. This begs the questions: in the presences of other targets of similar size, does the CNN have a good performance too? Is the CNN considering past entries, or the way fish move to make its predictions? Human observers can tell if it is a net or a fish because they have access to past and future entries (frames) and can recognize the difference in the way that a net and a fish move. Given only still images, humans would have a much harder time recognizing the blobs associated with nets.

Authors’ reply: Thank you for the comment, this is a good point. As you suggest, including temporal data, to capture motion, should definitely improve accuracy and is something we would like to do as future work, potentially with the use of a 3D CNN (to incorporate the temporal dimension) with minor adaptations, along with increasing the difficulty of the problem with greater variation in the training and test data. We have updated the manuscript to explain this (page 21). Here we have shown that with this dataset, even without temporal knowledge, we can still achieve sufficient accuracy with our CNN model. This suggests, on a like-for-like basis, where humans and CNN process still, non sequential images, our model will perform to a comparable degree of accuracy but the advantage is that it is much faster. 

Comment#9, Reviewer#1. Page 15. “Evaluation on DeepFish dataset”. Although an interesting exercise, I don't see how this relates to the counting on sonar images. These are two fundamentally different CNNs, especially their datasets. Underwater images (DeepFish) are visual data, the frequencies (colors) are directly associated with a bandwidth in the environment. Sonar images' gray scales are due to the mapping of the device, the data is sound mapped into a visual space, and thus, their bandwidth is dependent on the design of the device and the processing of the data. I am not sure that you can compare both. It is interesting to see that the framework does better that the original DeepFish architecture, but that does not necessarily means it would be better in sonar images.

Authors’ reply: Thank you for sharing your perspective on this analysis. We understand that this exercise does not show that our model would perform better on sonar images, the purpose of it was to show the generalisability of our model architecture and that its use does not need to be limited to sonar images. We hope it can be used in a wide range of aquatic research and perform well on a variety of image data. We have edited our discussion to make our goal clearer (page 19).

Comment#10, Reviewer#1. Page 17. “Its performance is notable when predicting samples containing higher numbers of fish and samples with substantial noise (Fig 5(a)), likely because by adding unlabelled data, we increase the number of samples that fall within these two challenging categories, which will disproportionately require more training data.” Is this speculation?

Authors’ reply: Yes, thank you. We have improved the wording in this excerpt to make this clear (page 18).

Comment#11, Reviewer#1. Page 18. “We also tested our proposed model on a recently published dataset, DeepFish”. See my comment on the "Evaluation of DeepFish dataset". The better performance of the framework here does not necessarily means a better performance with sonar images. The two datasets are fundamentally different.

Authors’ reply: Thank you again for highlighting this point. We have edited our discussion taking on board your feedback (page 19); please see our complete response to your previous Comment #9.

Comment#12, Reviewer#1. Page 18. “on both low-resolution, densely populated images”. I agree that ARIS acoustic images can be treated as low-resolution images. However, they are not the same as low-resolution visual images. They might be closer to low-resolution monochromatic images.

Authors’ reply: We have rephrased this term to low-resolution monochromatic images (page 19). 

Comment#13, Reviewer#1. Page 19. “in improving model predictions on unseen sonar data”. It is hard to find a study that does not support this. Only if the dataset is large enough, augmenting it does not help.

Authors’ reply: Thank you for pointing this out, we agree it is a widely known and used technique. We have updated the manuscript to make it more specific - i.e. both our studies started with a relatively small labelled dataset because very limited labelled data are available of this kind, but we were able to get to a good level of accuracy, partly due to the effectiveness of data augmentation (page 20).

Comment#14, Reviewer#1. Page 19. “to determine the minimum conditions of prey availability at which this traditional, century-old interaction can persist and remain resilient in face of the global trend of decline fisheries seen at local, regional and global scales”. But this was not the goal of the study

Authors’ reply: The reviewer is absolutely correct that this is not the goal of this study. In this closing section we aimed to provide the reader with the broader implications of our current technical study. We believe that by painting this bigger picture, we can better illustrate the usefulness of high-definition, localized fisheries data.

Comment#15, Reviewer#1. Page 19. “This is likely the cause of many errors in predictions”. Was there an attempt to quantify this?

Authors’ reply: No, but thanks for the suggestion. We first had two biologists work through a subsample together to reach a consensus on how to identify the number of mullet, but then only one annotated the entire dataset. Ideally we would have had multiple experts annotating the entire dataset, enabling us to not only detect errors, but also quantify and mitigate biases in ground-truth labeling (please see our reply to your comment #16). 

Comment#16, Reviewer#1. Page 20. “We make our data, code and pre-trained network available to benefit the advancement of other counting models via transfer learning”. There are models (such mean count and maximum count) used by marine biologist that work with visual data that could be applied here to give an estimate of the error in the Ground Truth, that, as the authors suggest, will be compounded into the CNN.

Authors’ reply: We appreciate these suggestions, and believe they will fit perfectly into a broader follow-up study. Thus, we now highlight the Reviewer’s points as directions for future work (pages 20-21). In addition, we are working on extending the dataset with more videos and also have multiple annotators, so we can reduce the noise by consensus after removing bad annotations based on inter-rater agreement metrics, as suggested by the recent literature [1-2].

[1] Kara, Yunus Emre, et al. "Actively estimating crowd annotation consensus." Journal of Artificial Intelligence Research 61 (2018): 363-405.

[2] Yu, Guoxian, et al. "Active multilabel crowd consensus." IEEE transactions on neural networks and learning systems 32.4 (2020): 1448-1459.

\f

Authors’ replies to the comments by Reviewer#2

Comment#1, Reviewer#2. Monitoring underwater populations is crucial to support the conservation of wild aquatic biota. The present manuscript proposes a deep learning method to estimate the number of fish in underwater images. The authors use a novel dataset of (low-resolution and site-specific) sonar images to train and test various density-based deep learning models. Finally, they argue the generalisability of their approach by testing the best performing model (the one with self-supervision and uncertainty regularization) on a contrasting dataset (high-resolution and heterogeneous-habitat) of underwater photos.

This is a really interesting contribution that fits well with the scope of Plos One, but I have a few minor issues concerning the analysis and some specific choices made by the authors. The manuscript might be considered acceptable after minor revision.

Authors’ reply: Thank you for the accurate summary highlighting the strengths of our work. We very much appreciate the positive feedback on the contributions of our work to the field, as well as the suggestions to improve the presentation of our manuscript. Please see below how we addressed all your comments (line numbers refer to the revised version with in-line tracked changes).

Comment#2, Reviewer#2. Referring to Equation 1, it is unclear to me how the count is extracted out of the predicted density map. Furthermore, I was not able to gain insights on this matter looking at the references [41-45,60]. From what I have understood, a pixel of the predicted density map contains a real number; how is the image predicted density normalized? Does the sum over spatial units provide a real number or an integer one (as it is reported in the estimated count in Figure 5 (c)-(f))?

Authors’ reply: Thank you for the attentive revision of our paper; excuse the lack of clarity in our text. Firstly, the point annotation to mark the location of a mullet for each labelled image will be interpreted as a “1” in that position in a corresponding array. A gaussian filter is passed over the entire array to create a density map, so the values of the original position of a given point annotation, and its surrounding area covered by the gaussian filter, will sum to 1 (hence normalized): the array will now consist of real numbers but not integers. Post filtering, any overlapping pixels from annotations from multiple mullets, will be added together - this will show as an area of greater density on the predicted density map. It is then possible to recover the predicted count of mullet from the density map simply by adding all the pixel values, as well as having a spatial representation to predict their location. We amended the commented excerpt to make these points explicit in the manuscript (page 8).

Comment#3, Reviewer#2. In subsection 'Ablation experiments', citing "All 9 trained models were tested on the 80-image test set for comparison and averaged across the 3 trials". The 3 trials are never mentioned before. I guess they refer probably to different simulation's run, each one with a diverse random seed. Maybe the authors can explicitly state it in the text. In this regard, do the results shown in Figure 5 (a), (b) refer to a specific trial or to an average of the 3?

Authors’ reply: Thank you for this feedback - we have updated the text as suggested to make this clearer (page 12). The Figures 5a and 5b provide the average of all trials to give an overview of our results. Figures 5c-f, where we show specific examples of predicted density maps, are from specific trials - here we have selected the median prediction for each sample image. 

Comment#4, Reviewer#2. The authors did not provide a reason for the choice of the hyperparameters used for training. Have they performed an hyperparameter optimization? Is the loss function of the models converging? Showing the learning curves (e.g. in Supplementary Materials) may provide useful informations.

Authors’ reply: Yes, we performed hyperparameter optimisation to choose model training hyperparameters such as the learning rate and the number of epochs to train our model for. We used an “early stopping procedure” to stop training when MAE of our validation data was at a minimum. We also experimented with training on increasing numbers of augmented images but found that accuracy plateaued beyond ~5,000 (hence using 5,322 in our ablation studies).We also performed hyperparameter optimisation on the coefficient of the balancing regularization as our model was not training with this being 1(as in [49]). Instead, the loss function converged best with a coefficient of 0.1 - the difference being our model inputs are whole images whereas [49] used patches of images. This means absolute differences between ground truth and prediction for a given input sample could be much higher in our results, resulting in the balancing function dominating the overall loss function and preventing training (please see page 12).

---

## [Decision Letter · Decision Letter 1]

18 Apr 2022

Deep learning with self-supervision and uncertainty regularization to count fish in underwater images

PONE-D-21-23858R1

Dear Dr. Cantor,

We’re pleased to inform you that your manuscript has been judged scientifically suitable for publication and will be formally accepted for publication once it meets all outstanding technical requirements.

Kind regards,

Tim Wilhelm Nattkemper

Academic Editor

PLOS ONE

Additional Editor Comments (optional):

Reviewers' comments:

Reviewer's Responses to Questions

**Comments to the Author**

1. If the authors have adequately addressed your comments raised in a previous round of review and you feel that this manuscript is now acceptable for publication, you may indicate that here to bypass the “Comments to the Author” section, enter your conflict of interest statement in the “Confidential to Editor” section, and submit your "Accept" recommendation.

Reviewer #1: All comments have been addressed

Reviewer #2: All comments have been addressed

2. Is the manuscript technically sound, and do the data support the conclusions?

Reviewer #1: Yes

Reviewer #2: Yes

3. Has the statistical analysis been performed appropriately and rigorously? 

Reviewer #1: Yes

Reviewer #2: Yes

4. Have the authors made all data underlying the findings in their manuscript fully available?

Reviewer #1: Yes

Reviewer #2: Yes

5. Is the manuscript presented in an intelligible fashion and written in standard English?

Reviewer #1: Yes

Reviewer #2: Yes

6. Review Comments to the Author

Reviewer #1: The authors addressed all my comments. I approve the manuscript for publication.

I commend the authors on working on this difficult subject where databases are hard to find and machine learning is usually ignored.

Reviewer #2: The manuscript satisfies PLOS ONE publication criteria and the authors have addressed all of my comments. I am happy to accept the paper for publication at this stage.

7. PLOS authors have the option to publish the peer review history of their article (what does this mean?). If published, this will include your full peer review and any attached files.

Reviewer #1: No

Reviewer #2: No

---

## [Editor Report · Acceptance letter]

25 Apr 2022

PONE-D-21-23858R1 

Deep learning with self-supervision and uncertainty regularization to count fish in underwater images 

Dear Dr. Cantor:

I'm pleased to inform you that your manuscript has been deemed suitable for publication in PLOS ONE. Congratulations! Your manuscript is now with our production department. 

Kind regards, 

on behalf of

Prof. Dr. Tim Wilhelm Nattkemper 

Academic Editor

PLOS ONE